# ADVERSARIALLY ROBUST CONFORMAL PREDICTION

**Asaf Gendler[1], Tsui-Wei Weng[2], Luca Daniel[2], Yaniv Romano[1]**
[1]*Department of Electrical and Computer Engineering, Technion - Israel Institute of Technology*
[2]*Department of Electrical Engineering and Computer Science, Massachusetts Institute of Technology*

## ABSTRACT

Conformal prediction is a model-agnostic tool for constructing prediction sets that are valid under the common i.i.d. assumption, which has been applied to quantify the prediction uncertainty of deep net classifiers. In this paper, we generalize this framework to the case where adversaries exist during inference time, under which the i.i.d. assumption is grossly violated. By combining conformal prediction with randomized smoothing, our proposed method forms a prediction set with finite-sample coverage guarantee that holds for any data distribution with $\ell_2$-norm bounded adversarial noise, generated by any adversarial attack algorithm. The core idea is to bound the Lipschitz constant of the non-conformity score by smoothing it with Gaussian noise and leverage this knowledge to account for the effect of the unknown adversarial perturbation. We demonstrate the necessity of our method in the adversarial setting and the validity of our theoretical guarantee on three widely used benchmark data sets: CIFAR10, CIFAR100, and ImageNet.

## 1 INTRODUCTION

Deep neural net classifiers have achieved tremendous accomplishments over the last several years. Nevertheless, the increased deployment of these algorithms in real-world applications, especially ones that can be life-threatening like autonomous driving, raises major concerns about their reliability (Heaven, 2019). To alleviate these issues, it is important to develop techniques that allow the users to assess the uncertainty in predictions obtained by complex classifiers, revealing their limitations.

Conformal prediction (Vovk et al., 2005) is a simple yet powerful tool for generating prediction sets whose size reflects the prediction uncertainty. Specifically, suppose we are given $n$ training examples $\{(X_i, Y_i)\}_{i=1}^{n}$ with feature vector $X_i \in \mathbb{R}^d$, discrete and unordered class label $Y_i \in \{1, 2, \ldots, L\} = \mathcal{Y}$, and any learning algorithm that aims at predicting the unknown $Y_{n+1}$ of a given test point $X_{n+1}$. Under the assumption that the training and test examples are sampled exchangeably—e.g., they may be drawn i.i.d.—from an unknown distribution $P_{XY}$, conformal prediction algorithms construct a distribution-free prediction set $\mathcal{C}(X_{n+1}) \subseteq \mathcal{Y}$ guaranteed to contain the test label $Y_{n+1}$ at any desired coverage probability $1 - \alpha \in (0, 1)$:

$$\mathbb{P}\left[Y_{n+1} \in \mathcal{C}(X_{n+1})\right] \geq 1 - \alpha. \tag{1}$$

For example, it is common to set the desired coverage level $1 - \alpha$ to be 90% or 95%. Note that the coverage probability $\mathbb{P}\left[Y_{n+1} \in \mathcal{C}(X_{n+1})\right]$ is marginal because it is taken over all the training examples $\{(X_i, Y_i)\}_{i=1}^{n}$ and the test point $X_{n+1}$. The key idea of conformal prediction is to fit a classifier on the training set and use this model to assign non-conformity scores for held-out data points. These scores reflect the prediction error of the underlying classifier, where, loosely speaking, a smaller prediction error would lead to the construction of smaller and more informative sets.

However, the sets constructed by the vanilla conformal method may not have the right coverage when the training and test points violate the exchangeability assumption (Cauchois et al., 2020; Gibbs & Candès, 2021; Tibshirani et al., 2019; Podkopaev & Ramdas, 2021; Guan & Tibshirani, 2019), which is hardly satisfied by real data in practice as distribution shift happens frequently (Koh et al., 2021). In particular, we consider the potential threat of adversarial attacks (Goodfellow et al., 2015; Szegedy et al., 2014; Carlini & Wagner, 2017)—carefully crafted human-imperceptible noise perturbations that drives the fitted model to err at test (inference) time. Such noise perturbations can introduce a large and arbitrary distribution shift that is extremely hard to estimate. In this setting, the

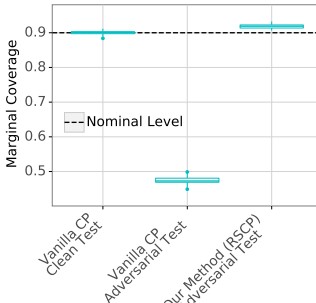 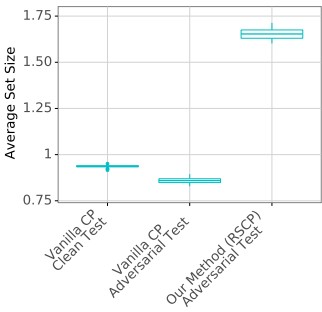

Figure 1: Marginal coverage (i.e., the percent of test labels that are included in the prediction sets) and average set-size obtained by vanilla conformal prediction (`Vanilla CP`) and our `RSCP` method, evaluated on the test set of CIFAR10. We use a VGG-19 classifier and set the nominal coverage level to be $(1 - \alpha) = 0.9$. See Section 5 and Supplementary Section S8 for more details.

prediction sets constructed by the vanilla conformal approach are often invalid, i.e., do not satisfy (1), as illustrated in Figure 1. Following that figure, while this method achieves the desired 90% coverage when applied to clean test data, the empirical coverage obtained when applying the same method on adversarial test examples falls dramatically below 90%, to about 30%.

Therefore, it is the main motivation of this work to construct prediction sets that are robust to adversarial attacks. We formalize this requirement as follows:

$$\mathbb{P}\left[Y_{n+1} \in \mathcal{C}_\delta(\tilde{X}_{n+1})\right] \geq 1 - \alpha, \tag{2}$$

where $\tilde{X}_{n+1} = X_{n+1} + \epsilon$ is the test adversarial example, and $\|\epsilon\|_2 \leq \delta$ is a norm-bounded adversarial perturbation. Note that we use the notation $\mathcal{C}_\delta$ to distinguish between the new setting from the exchangeability case for which $\delta = 0$. Crucially, we require (2) to hold in finite samples, for any $P_{XY}$, any adversarial perturbation of magnitude bounded by $\delta$, and regardless of the choice or accuracy of the underlying classifier. At the same time, we wish $C_\delta$ to be as small as possible.

To realize (2) with the desired coverage, we propose to combine randomized smoothing (Duchi et al., 2012; Cohen et al., 2019; Salman et al., 2019) with the vanilla conformal prediction procedure, and hence we name our technique **R**andomly **S**moothed **C**onformal **P**rediction (`RSCP`). Randomized smoothing allows us to bound the Lipschitz constant of any non-conformity score function by convolving it with the Gaussian distribution function. Leveraging this bound, we show how to modify conformal prediction and rigorously construct prediction sets that account for the adversarial perturbation. Figure 1 illustrates that our proposed `RSCP` approach successfully attains the desired coverage level whereas the vanilla conformal method fails. Observe that `RSCP` constructs slightly larger prediction sets, reflecting the increased uncertainty induced by the adversarial noise.

The contributions of this paper are two-fold: (i) We propose, for the first time, a new conformal prediction method that can account for the potential adversarial threats during inference time. Our `RSCP` method, described in Section 3, is *model-agnostic* in that it can work with any classifier and non-conformity score, *scalable* since the smoothing can be done by Monte Carlo integration with many i.i.d. Gaussian noise realizations, and *robust* against any $\ell_2$-norm bounded adversarial attack. (ii) We prove that the prediction sets constructed by `RSCP` are valid in the sense of (2), and, in Section 5, support this theoretical result with numerical experiments on CIFAR10, CIFAR100, and ImageNet data sets.

## 2 CONFORMAL PREDICTION

Since the focus of this paper is on how to adapt the vanilla conformal prediction method to the adversarial setting, in this section, we give background on conformal prediction. While we focus here on classification problems, the method of conformal prediction can also be applied to regression tasks. The vanilla conformal prediction can be divided into two categories: the split conformal prediction

and the full conformal prediction. The first one involves data splitting while the second one constructs valid sets without splitting the data but at the cost of a significant increase in computational complexity at test time (Papadopoulos et al., 2002; Papadopoulos, 2008; Vovk et al., 2005). To avoid prohibitive computation complexity, in this paper we focus only on split conformal prediction.

This method starts by splitting the training data into two disjoint subsets: a proper training set $\mathcal{I}_{\text{tr}} \subseteq \{1, \ldots, n\}$ and a calibration set $\mathcal{I}_{\text{cal}} = \{1, \ldots, n\} \setminus \mathcal{I}_{\text{tr}}$. Then, a classifier $\hat{f}(x) \in [0, 1]^L$ is fit to the proper training set, estimating the conditional class probabilities $\mathbb{P}[Y = y \mid X = x]$ for all $y \in \mathcal{Y}$. In the case of deep net classifiers, which is the focus of this work, this may be the output of the softmax layer. Next, we compute a non-conformity score $S_i = S(X_i, Y_i) \in \mathbb{R}$ for each calibration point $\{(X_i, Y_i)\}_{i \in \mathcal{I}_{\text{cal}}}$. This score expresses how well the model prediction $\hat{f}(X)$ is aligned with the true label $Y$, where a lower score implies better alignment. For example, the score from Vovk et al. (2005); Lei et al. (2013) is given by

$$S(x, y) = 1 - \hat{f}(x)_y, \tag{3}$$

where $\hat{f}(x)_y \in [0, 1]$ is the $y$th entry in the vector $\hat{f}(x)$. Another example is the score proposed by Romano et al. (2020b), which can be expressed as

$$S(x, y) = \sum_{y' \in \mathcal{Y}} \hat{f}(x)_{y'} \, \mathbb{I}\left\{\hat{f}(x)_{y'} > \hat{f}(x)_y\right\} + \hat{f}(x)_y \cdot u, \tag{4}$$

where $\mathbb{I}$ is the indicator function and $u$ is a random variable distributed uniformly over the segment $[0, 1]$. We refer to the score from (3) as HPS as it was shown to construct *homogeneous prediction sets*. Analogously, we refer to (4) as APS since it tends to yield *adaptive prediction sets* that reflect better the underlying uncertainty across sub-populations; see Romano et al. (2020b) for more details. Given the desired coverage level $1 - \alpha$, the prediction set for a new test point $X_{n+1}$ is formulated as

$$\mathcal{C}(X_{n+1}) = \left\{y \in \mathcal{Y} : S(X_{n+1}, y) \leq Q_{1-\alpha}\left(\{S_i\}_{i \in \mathcal{I}_{\text{cal}}}\right)\right\}, \tag{5}$$

where

$$Q_{1-\alpha}\left(\{S_i\}_{i \in \mathcal{I}_{\text{cal}}}\right) := \text{ the } (1 - \alpha)\left(1 + \frac{1}{1 + |\mathcal{I}_{\text{cal}}|}\right) \text{th empirical quantile of } \{S_i\}_{i \in \mathcal{I}_{\text{cal}}} \tag{6}$$

is the score positioned $\lceil (n+1)(1-\alpha) \rceil$ in the sorted array of calibration scores $S_i, i \in \mathcal{I}_{\text{cal}}$. In plain words, in (5) we sweep over all possible labels $y \in \mathcal{Y}$ and include in $\mathcal{C}(X_{n+1})$ the 'guessed' labels $y$ whose scores $S(X_{n+1}, y)$ are smaller than most of the calibration scores $S(X_i, Y_i)$. Since the calibration and test points are drawn exchangeably from $P_{XY}$ and $\hat{f}$ is fixed, the score $S(X_{n+1}, y)$ for the guess $y = Y_{n+1}$ can fall anywhere in the sorted array of calibration scores with equal probability. This property guarantees that the prediction set (5) satisfies (1); see Vovk et al. (2005).

## 3 RANDOMLY SMOOTHED CONFORMAL PREDICTION

In this section, we introduce our proposed RSCP framework for constructing prediction sets that are valid in the adversarial regime. Recall that an adversarial attack can lead to a significant distributional shift between the clean calibration points and the corrupted test example $\tilde{X}_{n+1} = X_{n+1} + \epsilon$, thus violating the fundamental exchangeability assumption of the split conformal procedure. Focusing on the guess $y = Y_{n+1}$, an effective attack would result in a larger non-conformity score for the corrupted test point $S(\tilde{X}_{n+1}, y)$ compared to that of the clean input $S(X_{n+1}, y)$. Therefore, a naive comparison of $S(\tilde{X}_{n+1}, y)$ to the same threshold $Q_{1-\alpha}$ from (6), which neglects the increased uncertainty caused by the adversarial perturbation, will result in a prediction set that may not achieve the desired coverage, as already illustrated in Figure 1. To address this, we should compare the test score to an *inflated* threshold, larger than $Q_{1-\alpha}$, which rigorously accounts for the effect of the adversarial noise. This is the core idea behind our proposal described in detail below; see Figure 2.

### 3.1 ADVERSARIALLY ROBUST CALIBRATION

Suppose we are given a non-conformity score function $\tilde{S}$ for which we can bound by how much its value could be increased due to the adversarial noise $\|\epsilon\|_2 \leq \delta$ added to $X_{n+1}$. Formally, we require the score $\tilde{S}$ to satisfy the following relation:

$$\tilde{S}(\tilde{X}_{n+1}, y) \leq \tilde{S}(X_{n+1}, y) + M_\delta, \quad \forall y \in \mathcal{Y}, \tag{7}$$

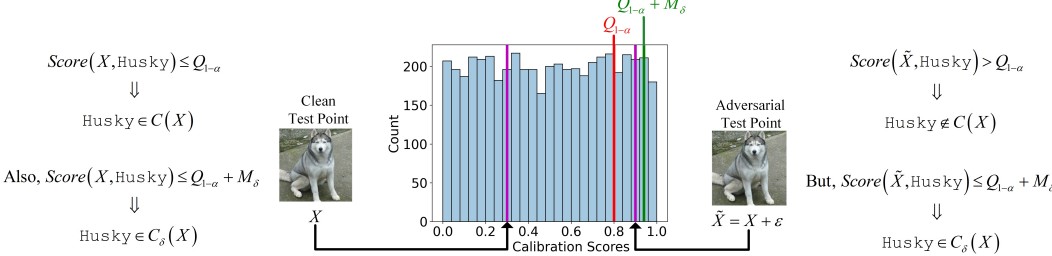

Figure 2: Schematic demonstration of the effect of an adversarial noise on vanilla split conformal prediction, along with our proposed solution. The vanilla conformal set $\mathcal{C}(X)$ and our RSCP set $\mathcal{C}_\delta(X)$ are generated as described in (5) and (8), respectively. The black arrows point to the values of the non-conformity scores that correspond to the clean (left) and adversarial (right) test points, which are further marked by the vertical purple lines.

where $M_\delta \geq 0$ is a constant that is a function of $\delta$, such that $M_{\delta_1} \geq M_{\delta_2}$ for $\delta_1 \geq \delta_2$ and $M_\delta = 0$ for $\delta = 0$. In essence, we would like (7) to hold for the smallest possible $M_\delta$. We denote this score function by $\tilde{S}$ to emphasize that it must satisfy (7), distinguishing it from existing non-conformity scores $S$, e.g., (3)–(4); see Section 3.2 for a concrete and very general framework for designing $\tilde{S}$ for which the constant $M_\delta$ can be easily derived. Importantly, $M_\delta$ serves as a bridge between the observed score $\tilde{S}(\tilde{X}_{n+1}, y)$ and the unobserved one $\tilde{S}(X_{n+1}, y)$ for a fixed $y \in \mathcal{Y}$. Leveraging this property, we propose to construct a prediction set robust to a norm-bounded adversarial attack by applying the following decision rule:

$$\mathcal{C}_\delta(\tilde{X}_{n+1}) = \{y \in \mathcal{Y} : \tilde{S}(\tilde{X}_{n+1}, y) \leq Q_{1-\alpha}(\{\tilde{S}_i\}_{i \in \mathcal{I}_{\text{cal}}}) + M_\delta\}, \tag{8}$$

where $\tilde{S}_i = \tilde{S}(X_i, Y_i)$. In contrast to the vanilla split conformal approach (5), the prediction set defined above is generated by comparing the test score to an inflated threshold $Q_{1-\alpha} + M_\delta$, as illustrated in Figure 2. Notice that the level of inflation is affected by the magnitude of the adversarial perturbation as well as the robustness of $\tilde{S}$, i.e., the value of $M_\delta$. A larger perturbation implies larger inflation, and a more robust score $\tilde{S}$ implies smaller inflation. The theorem below states that the constructed prediction set (8) is guaranteed to contain the unknown target label $Y_{n+1}$ with a probability of at least $1 - \alpha$, for any distribution $P_{XY}$, sample size $n$, score function $\tilde{S}$ that satisfies (7), and adversarial perturbation of magnitude $\delta$ generated by any attack algorithm. The proof of this and all other results can be found in Section S1 of the Supplementary Material.

**Theorem 1.** *Assume that the samples $\{(X_i, Y_i)\}_{i=1}^{n+1}$ are drawn exchangeably from some unknown distribution $P_{XY}$. Let $\tilde{X}_{n+1} = X_{n+1} + \epsilon$ be a clean test example $X_{n+1}$ with an additive corruption of an $\ell_2$-norm bounded adversarial noise $\|\epsilon\|_2 \leq \delta$. Then, the prediction set $\mathcal{C}_\delta(\tilde{X}_{n+1})$ defined in* (8) *satisfies*

$$\mathbb{P}\left[Y_{n+1} \in \mathcal{C}_\delta(\tilde{X}_{n+1})\right] \geq 1 - \alpha.$$

Before presenting our framework to construct scores that rigorously satisfy (7), we pause to prove a lower bound on the coverage that the vanilla split conformal could attain in the adversarial setting.

**Theorem 2.** *Under the assumptions of Theorem 1, the prediction set $\mathcal{C}(\tilde{X}_{n+1})$ defined in* (5)*, applied with $\tilde{S}$ in place of $S$, satisfies*

$$\mathbb{P}\left[Y_{n+1} \in \mathcal{C}(\tilde{X}_{n+1})\right] \geq \tau,$$

*where*

$$\tau = \max\left\{\tau' : Q_{\tau'}\left(\{\tilde{S}_i\}_{i \in \mathcal{I}_{\text{cal}}}\right) \leq Q_{1-\alpha}\left(\{\tilde{S}_i\}_{i \in \mathcal{I}_{\text{cal}}}\right) - M_\delta\right\}. \tag{9}$$

Note that $\tau$ can be simply computed by running a grid search on the sorted array of calibration scores. It is important to observe that in contrast to Theorem 1, which guarantees at least $1 - \alpha$ coverage for any user-specified level $\alpha$, the worst coverage level $\tau$ of the vanilla split conformal is not controlled explicitly, i.e., $\tau$ is known only after looking at the training and calibration data. Observe also that,

by construction, $\tau \leq 1 - \alpha$ (see Supplementary Section S1) and equality is met in a special case where $\delta = 0$, i.e., no attack is performed. In this special case, $M_\delta = 0$ by definition, and both Theorem 1 and 2 converge to the classic coverage guarantee of the vanilla conformal algorithm.

## 3.2 THE RANDOMLY SMOOTHED SCORE

To apply the robust calibration procedure presented above, we need to have access to a constant $M_\delta$ that satisfies (7) for a given non-conformity score function. In general, this constant is unknown for modern deep neural net classifiers as well as for complex non-conformity scores, such as the one defined in (4). In what follows, we present a novel framework that takes inspiration from the literature on randomized smoothing (Duchi et al., 2012; Cohen et al., 2019; Salman et al., 2019), offering a theoretically grounded mechanism to bound the Lipschitz constant of any scoring function derived from any black-box classifier. Concretely, we suggest constructing a new score $\tilde{S}$ from the base score $S$ as follows:

$$\tilde{S}(x, y) = \Phi^{-1}\left(\mathbb{E}_{v \sim \mathcal{N}(0_d, \sigma^2 I_d)}\left[S(x + v, y)\right]\right), \tag{10}$$

where $\tilde{S}$ is a "smoothed" version of the base score $S$, obtained by averaging the value of $S(x + v, y)$ over many independent samples $v \sim \mathcal{N}(0_d, \sigma^2 I_d)$ drawn from the multivariate normal distribution, and then applying $\Phi^{-1}$, the inverse of the cumulative distribution function (CDF) of the standard normal distribution. The hyper-parameter $\sigma$ controls the level of smoothing, where a larger $\sigma$ implies stronger smoothing. (In Section 5.1, Figure 3, and Supplementary Section S5.1 we discuss the effect of this parameter on the statistical efficiency of the overall calibration procedure.) Assuming that the base score function $S(x, y) \in [0, 1]$ for any given $(x, y)$, which holds for most non-conformity scores in multi-class classification problems, we can invoke a well-known result from the randomized smoothing literature (Salman et al., 2019, Lemma 2) and get the following upper-bound for the smoothed score:

$$\tilde{S}(\tilde{X}_{n+1}, y) \leq \tilde{S}(X_{n+1}, y) + \frac{\|\epsilon\|_2}{\sigma} \leq \tilde{S}(X_{n+1}, y) + \frac{\delta}{\sigma}. \tag{11}$$

Importantly, (11) holds for every $y \in \mathcal{Y}$, hence rigorously satisfying the relation (7) for the choice of $M_\delta = \delta / \sigma$. Armed with $\tilde{S}$, we can invoke Theorems 1- 2 and construct an adversarially robust prediction set by following (8) or lower bound the coverage of the vanilla split conformal procedure.

**Corollary 1.** *Let $\tilde{S}$ be the randomly smoothed score function* (10) *and set $M_\delta = \delta / \sigma$. The prediction set defined in* (8) *satisfies $\mathbb{P}\left[Y_{n+1} \in \mathcal{C}_\delta(\tilde{X}_{n+1})\right] \geq 1 - \alpha$, and the worst coverage of* (5) *is given by $\mathbb{P}\left[Y_{n+1} \in \mathcal{C}(\tilde{X}_{n+1})\right] \geq \tau$, where $\tau = \max\left\{\tau' : Q_{\tau'}\left(\{\tilde{S}_i\}_{i \in \mathcal{I}_{\mathrm{cal}}}\right) \leq Q_{1-\alpha}\left(\{\tilde{S}_i\}_{i \in \mathcal{I}_{\mathrm{cal}}}\right) - \frac{\delta}{\sigma}\right\}$.*

The above leads to two possible use-cases of our proposal. In the first, we can implement our randomly smoothed conformal prediction procedure (`RSCP`) (8) and construct valid prediction sets at any desired level, bearing in mind that these sets are likely to be larger due to the increased uncertainty induced by the adversarial perturbation. In the second case, we can deploy the vanilla split conformal algorithm with a smoothed version of the base score, providing important information about the worst coverage that might be obtained under an adversarial attack.

In practice, the smoothed score cannot be evaluated explicitly as it requires to compute the expectation in (10). However, this smoothed score can be easily estimated by averaging over many i.i.d. Gaussian noise realizations in a Monte Carlo fashion (Cohen et al., 2019, Section 3.2), as described in Algorithm 1; see Section 5.1 and Supplementary Section S5.2 for more details about the effect of this approximation on the coverage and set-size. Here, an interesting future direction is to develop a high probability bound for the smoothed score approximation error and integrate it with our method.

## 4 RELATED WORK

### 4.1 CONFORMAL PREDICTION UNDER DISTRIBUTION SHIFT

Previous work offers several generalizations of the vanilla conformal prediction framework to the case of distribution shift between the training and the test points, however, none of them addresses

---

**Algorithm 1** `RSCP`: Randomly Smoothed Conformal Prediction

---

**Input**: Data $\{(X_i, Y_i)\}_{i=1}^n$; a test point $\tilde{X}_{n+1}$ corrupted by adversarial noise of energy $\delta$; learning algorithm $\mathcal{A}$; a base non-conformity score $S$; and coverage level $1 - \alpha$.

  1: Split the training data into 2 disjoint subsets $\mathcal{I}_{\text{tr}}$ and $\mathcal{I}_{\text{cal}}$.
  2: Train a classifier using $\mathcal{A}$ on all samples from $\mathcal{I}_{\text{tr}}$, i.e., $\hat{f}(X) \leftarrow \mathcal{A}\left(\{(X_i, Y_i)\}_{i \in \mathcal{I}_{\text{tr}}}\right)$.
  3: Draw $v_{ij} \overset{i.i.d.}{\sim} \mathcal{N}\left(0_d, \sigma^2 I_d\right)$, where $i \in \mathcal{I}_{\text{cal}}$ and $j = 1, \ldots, n_s$, and compute:

$$\hat{S}_i = \hat{S}(X_i, Y_i) = \Phi^{-1}\left(\frac{1}{n_s}\sum_{j=1}^{n_s} S(X_i + v_{ij}, Y_i)\right), \quad \text{for all } i \in \mathcal{I}_{\text{cal}}. \tag{12}$$

  4: Compute $Q_{1-\alpha}(\{\hat{S}_i\}_{i \in \mathcal{I}_{\text{cal}}}) :=$ the $(1-\alpha)\left(1 + \frac{1}{1 + |\mathcal{I}_{\text{cal}}|}\right)$ th empirical quantile of $\{\hat{S}_i\}_{i \in \mathcal{I}_{\text{cal}}}$.
  5: Construct a prediction set for $\tilde{X}_{n+1}$:

$$\mathcal{C}_\delta(\tilde{X}_{n+1}) = \left\{y \in \mathcal{Y} : \hat{S}(\tilde{X}_{n+1}, y) \leq Q_{1-\alpha}(\{\hat{S}_i\}_{i \in \mathcal{I}_{\text{cal}}}) + \frac{\delta}{\sigma}\right\}.$$

**Output**: A prediction set $\mathcal{C}_\delta(\tilde{X}_{n+1})$ for the unobserved test label $Y_{n+1}$.

---

the adversarial setting specifically. Cauchois et al. (2020) suggested a method to construct valid sets when the test distribution of the non-conformity scores is in an $f$-divergence ball around their training distribution. While this approach shares the idea of inflating the value of $Q_{1-\alpha}$, the $f$-divergence measure is notoriously difficult to estimate in practice (Nguyen et al., 2010; Tsybakov, 2009) and thus less applicable compared to our approach, which focuses on the addition of bounded $\ell_2$ adversarial noise (see Supplementary Section S12 for a comparison between the methods). Gibbs & Candès (2021) generalized the vanilla conformal prediction approach to an *online* setting where the data generating distribution varies over time by modifying the threshold adaptively as new *labeled* points are being observed, such that the *long-term* empirical coverage frequency would be at least $1 - \alpha$. This is of course different from our approach that works *offline* and assumes the same data generating process but with the addition of a malicious small perturbation induced by an attacker. Tibshirani et al. (2019) studied the case of covariate shift and offered a weighted version of conformal prediction to address it. In this setting, the shift is only in $P_X$ while $P_{Y|X}$ remains intact. Podkopaev & Ramdas (2021) modified this method to tackle the label-shift case, in which $P_{X|Y}$ remains the same but $P_Y$ varies between train and test time. Both approaches differ from ours, which handles a full distributional shift induced by the adversarial perturbation. Lastly, Hechtlinger et al. (2018) suggested a non-conformity score that empirically tends to be more robust to adversarial attacks, replacing the classifier's class probability estimates of $Y \mid X$ with a density estimator that approximates the conditional distribution of $X \mid Y$. In contrast to our `RSCP` approach, this method is not supported by a theoretical guarantee for attaining the desired coverage in the adversarial setting.

## 4.2 Certified adversarial robustness

A different but related line of research is known as 'robustness certification'. Here, the goal is to provide an $\ell_p$ radius around a test point, in which no adversarial perturbations exist, i.e, the predicted label provably remains the same inside this radius. Currently, one of the mainstream approaches in robustness certification is through randomized smoothing, which was proposed in (Lecuyer et al., 2019; Li et al., 2019; Cohen et al., 2019) to provide an analytic form of robustness certificate. More formally, given a base classifier $\hat{f}(x)$, one can construct a smoothed classifier according to the following decision rule:

$$\hat{y} = \text{argmax}_y \ \mathbb{P}[\hat{f}_y(x + v)], \ v \sim \mathcal{N}(0_d, \sigma^2 I_d). \tag{13}$$

In contrast to the base classifier, the smoothed classifier can be supported by a rigorous $\ell_2$ certification radius around the test point $X_{n+1} = x$. It is self-evident that our work is different from robustness certification, as we are not aiming to deliver a robustness certificate for a given classifier over some test point; instead, we construct a valid prediction set whose size can vary with $x$. However, both works make use of the randomized smoothing technique, sharing related set of ideas.

For example, practical evidence shows that for the smoothed model (13) to correctly classify the test point $X_{n+1}$, the base classifier $\hat{f}$ needs to consistently label the noisy sample $\mathcal{N}\left(X_{n+1}, \sigma^2 I\right)$ as $Y_{n+1}$ (Cohen et al., 2019, Section 3.3). Training the base classifier without accounting for the future smoothing operation would not necessarily satisfy this requirement, and, as a consequence, would lead to a very small certification radius. We note that this observation is also relevant to our setting, where we found that using a classifier that is not robust to Gaussian perturbations results in prediction sets that are often large and thus uninformative. More details are provided in Section S4 of Supplementary Material. One way to improve the certified radius is by training the base classifier on the labeled training pairs $\{(\mathcal{N}\left(X_i, \sigma^2 I\right), Y_i)\}_i$, $i \in \mathcal{I}_{\text{tr}}$ as in Lecuyer et al. (2019), where $\sigma$ is equal to the one used for smoothing. Later, Salman et al. (2019) further improved the robustness guarantee through adversarial training, where the adversary generates a bounded perturbation via the method of SMOOTHADV—an adversarial attack designed specifically against smoothed classifiers. A more recent work by Salman et al. (2020) suggested adding a denoising pre-processing layer preceding the classifier, to remove the Gaussian noise used in randomized smoothing. In our experiments, we find that the training strategy proposed in Lecuyer et al. (2019) works well in practice and results in relatively small prediction sets, although a better training scheme—such as the one from Salman et al. (2019) or Salman et al. (2020)—could be easily combined with our method and may further improve the overall statistical efficiency.

## 5    EXPERIMENTS

In this section, we evaluate the performance of our proposed methods on three benchmark image classification data sets: CIFAR10, CIFAR100 (Krizhevsky, 2009), and ImageNet ILSVRC2012 (Deng et al., 2009), which are described in Supplementary Section S2. We evaluate our methods as follows. First, we fit a model on the entire training set. Then, we split the remaining data into two equally sized disjoint subsets, one is used for calibration and the second for testing. The calibration is done using the two base non-conformity scores mentioned in Section 2, i.e., HPS (3) and APS (4). Next, we attack each point in the test set by adding to it an adversarial perturbation whose $\ell_2$-norm is bounded by $\delta$.[1] Lastly, for each adversarial test point, we construct a prediction set using three different conformal methods (described below), asking for a nominal coverage level of $1 - \alpha$. Given the constructed sets, we test the validity of each method by reporting the marginal coverage rate, and the statistical efficiency by computing the average size of the prediction sets. We repeat this process 50 times by randomly assigning data points to form the calibration and test sets.

In our experiments, we compare three different approaches: (i) vanilla split conformal prediction implemented with the two base-scores (HPS and APS), denoted by Vanilla CP; (ii) vanilla split conformal prediction with our proposed smoothed version of the base scores, denoted by CP+SS; and (iii) the proposed randomly smoothed conformal prediction (see Algorithm 1), denoted by RSCP. Our experiments show that Vanilla CP constructs invalid prediction sets whose marginal coverage is way below the desired level. The smoothed score improves robustness and coverage but still does not achieve the nominal coverage rate. The prediction sets constructed by our RSCP approach achieve the desired coverage level, aligning with our theoretical guarantees.

### 5.1    IMPLEMENTATION DETAILS

**Models**    For all data sets, we choose ResNet (He et al., 2016) to be the base architecture of the deep net classifier and fit two different models for each data set: one on clean training points and the second on points that are augmented with Gaussian noise of standard deviation $\sigma$ that is equal to the smoothing parameter from (10). The former is used to explore the effect of adversarial examples on the validity of Vanilla CP. The latter training strategy improves the robustness of the classifier and, as a consequence, reduces the size of the prediction sets constructed by CP+SS and RSCP methods, as discussed in Section 4.2 and Supplementary Section S4. For CIFAR10 and ImageNet data sets, we use the pre-trained ResNet-110 and ResNet-50 models from Cohen et al. (2019), and for CIFAR100 we fit our own ResNet-110 model as the latter data set is not studied by Cohen et al. (2019), please see Supplementary Section S3 for additional details on the training procedure. See also Supplementary Section S8 for similar experiments in which we choose DenseNet (Iandola et al., 2014) and VGG (Simonyan & Zisserman, 2014) to be the base architectures.

---

[1]See Supplementary Section S9 for additional experiments evaluated on clean data points.

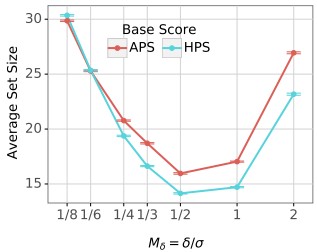

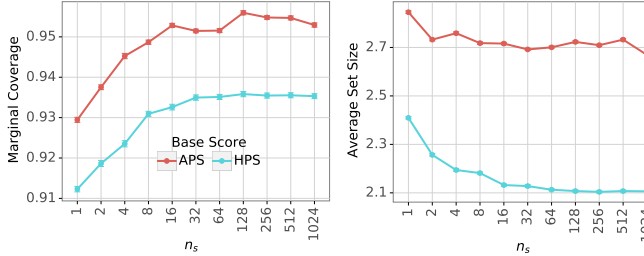

Figure 3: The effect of $\sigma$ on the size of RSCP's prediction sets. The results correspond to the CIFAR100, with $\delta = 0.125$ and $n_s = 256$.

Figure 4: The effect of the number of noise realizations $n_s$ (used to estimate the smoothed score) on the marginal coverage (left) and size (right) of prediction sets constructed by RSCP. The results correspond to the CIFAR10 data set, where we set $\delta$ and $\sigma$ to be equal to $0.125$ and $0.25$, respectively.

**Attack algorithm**   For Vanilla CP, we generate adversarial noise using the projected gradient descent (PGD) attack (Madry et al., 2018), which is one of the strongest white-box attacks known in the literature to construct $\ell_2$ bounded adversarial examples. For the other two methods that make use of randomized smoothing (CP+SS and RSCP), we apply SMOOTHADV (Salman et al., 2019)—a variation of PGD that is tailored for smoothed classifiers. For CIFAR10 and CIFAR100, we attack the test points with an adversarial perturbation of energy $\delta = 0.125$, and for ImageNet we use a larger perturbation of magnitude $\delta = 0.25$. Both choices lead to a significant reduction in the test accuracy of the base classifier, as summarized in Supplementary Section S6.

**Choosing $\sigma$ for smoothing**   Recall Corollary 1 and observe that the coverage guarantee holds for any choice of the smoothing level $\sigma$. This parameter is intimately connected to the accuracy/robustness trade-off (Cohen et al., 2019, Section 4): a large $\sigma$ will reduce the Lipschitz constant $M_\delta = \delta/\sigma$, however, may also reduce the accuracy of the base classifier (even if fitted on noisy samples). Similar to previous work on randomized smoothing (Cohen et al., 2019; Salman et al., 2019), we treat $\sigma$ as a hyper-parameter of our method which can be tuned, for example, by choosing $\sigma$ that produces the smallest sets by cross-validation.[2] In practice, we find that setting $\sigma = 2\delta$ in (10) yields informative prediction sets, as illustrated in Figure 3. This figure also demonstrates the accuracy/robustness trade-off discussed above: the size of the prediction sets reduces for $1/8 \leq M_\delta \leq 1/2$, and increases for $M_\delta > 1/2$. See Supplementary Section S5.1 for additional experiments that study the effect of $\sigma$ on the marginal coverage and average set-size for all data sets.

**Choosing the number of Monte Carlo iterations**   Following Algorithm 1, the smoothed score is evaluated by replacing the expectation (10) with the empirical mean (12), averaging over $n_s$ i.i.d. Gaussian noise samples. This is similar to the approach Cohen et al. (2019, Section 3.2) used for approximating the smoothed classifier defined in (13). In our experiments, we find that our method is fairly robust to the choice of $n_s$, and using more than $n_s = 256$ realizations barely affects the marginal coverage and average set-size. This phenomenon is demonstrated in Figure 4 for CIFAR10; see Supplementary Section S5.2 for additional experiments that illustrate the influence of $n_s$ on CIFAR100 and ImageNet. Therefore, in our experiments we set $n_s = 256$ both for CIFAR10 and CIFAR100, and $n_s = 64$ for ImageNet due to limited GPU memory footprint; see Supplementary Section S7 for a discussion about the computational complexity including the running time of RSCP.

## 5.2   RESULTS

Figure 5 presents the marginal coverage obtained by Vanilla CP, CP+SS, and RSCP for the three data sets. As can be seen, Vanilla CP fails to construct valid prediction sets in the presence of adversarial noise. For instance, consider the choice of HPS as the non-conformity score, and observe that the average coverage for CIFAR10, CIFAR100, and ImageNet is $27.27\%$, $29.24\%$, and $38.02\%$, respectively; all are much smaller than the desired $90\%$ level, emphasizing the necessity of

---

[2]Note that our objective is different from that of certified robustness that seeks for $\sigma$ that leads to a large radius in which the prediction is certifiably robust.

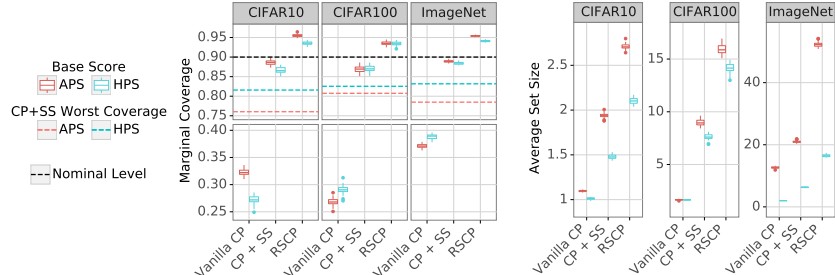

Figure 5: Marginal coverage (left) and average set-size (right) obtained by different conformal methods. The target coverage level is $90\%$.

our proposal. When using the same vanilla split conformal procedure, however with the smoothed version of the base non-conformity score (i.e., `CP+SS`), we get a major improvement in terms of coverage. Here, for the smoothed version of the HPS score, the average coverage levels obtained for CIFAR10, CIFAR100, and ImageNet are closer to $90\%$ and are equal to $86.78\%$, $87.10\%$, and $88.12\%$, respectively. This serves as evidence that the new smoothed score is more robust to adversarial attacks. Moreover, since we can derive the Lipschitz constant of the smoothed score, we can invoke the worst coverage bound from Theorem 2. As illustrated, the (average) worst coverage is indeed below the empirical coverage level obtained by `CP+SS`. While this lower bound is close to the empirical one (for both scores), there exists a gap between the two that can be possibly reduced by designing a better algorithm tailored to attack the smoothed score. By contrast, our `RSCP` method generates valid prediction sets whose coverage is slightly above the nominal level, supporting our theoretical guarantee.

Figure 5 also presents the average set-size constructed by all three methods. Observe that `CP+SS` generates larger sets compared to `Vanilla CP` since it relies on a more robust classifier that leads to better coverage at cost of lower accuracy; see Supplementary Table S1. Observe also that `RSCP` constructs larger sets than those of `CP+SS` due to the inflated threshold used in the calibration, accounting for the effect of the adversarial noise. In addition, the sets obtained by the APS score are larger than those of HPS. This can be explained by (i) the coverage plots: the marginal coverage obtained when using the smoothed APS score tends to be larger than that of the smoothed HPS; and (ii) it is known that the APS score tends to construct larger sets even in the noiseless case since it aims at constructing sets of $1 - \alpha$ coverage across different sub-populations; see Supplementary Section S11 for an experiment that illustrates this as well as Romano et al. (2020b).

# 6 CONCLUSIONS AND FUTURE WORK

In this paper, we generalized the technique of conformal prediction to the adversarial setting and introduced a model agnostic, scalable, and simple method for constructing valid prediction sets to multi-class classification problems, accounting for the adversarial noise. Our experiments show that our `RSCP` method is most effective when combined with robust classifiers, e.g., ones that are fitted by augmenting Gaussian noise to the training data. However, this often leads to a degradation in test accuracy, a limitation that is not unique to our work. In fact, this is a matter of ongoing research in the robustness certification literature, and note that any new development in that front (e.g., Salman et al. (2020)) could be easily integrated with our method as demonstrated in Supplementary Section S10. An exciting future direction could be to extend the guarantees we provided beyond the assumption that the adversarial noise is $\ell_2$-norm bounded and consider different smoothing and calibration techniques that can handle the $\ell_0$, $\ell_1$, and $\ell_\infty$ cases (Lee et al., 2019; Teng et al., 2019; Zhang et al., 2019). Other directions could be to explore the use of our method to improve robustness to adversarial attacks in regression problems and to provide conditional validity guarantees of the style suggested by Vovk (2012).

ACKNOWLEDGMENTS

A.G. and Y.R. were partially supported by the 2020 Zuckerman Fund, Technion Center for Machine Learning and Intelligent Systems (MLIS Grant No. 2029967), and the ISRAEL SCIENCE FOUNDATION (grant No. 729/21). Y.R. thanks the Career Advancement Fellowship, Technion, for providing research support. Y.R also thank Stephen Bates for his comments on an earlier version of this manuscript.

ETHICS STATEMENT

The ability to communicate the predictive uncertainty of any classifier by constructing prediction sets of valid coverage is crucial to assess the performance of machine learning algorithms (Holland, 2020), improve their reliability, and avoid unwanted discrimination (Romano et al., 2020a). Our proposed method further enriches the existing uncertainty quantification toolbox, by providing a rigorous form of robustness to adversarial perturbations. For example, one can promote fairness in high-stakes applications (in addition to robustness) by integrating RSCP within the framework of equalized coverage (Romano et al., 2020a). Here, we stress that the validity of our methods relies on the exchangeability assumption between the *clean* training and test points, and on the knowledge of the $\ell_2$-bound of the attack. If one of these assumptions is violated, the sets that RSCP constructs may not have the right coverage. Therefore, it is recommended to assess whether the above assumptions are reasonable when using the proposed methods.

REPRODUCIBILITY STATEMENT

A Python package that implements our methods and code to reproduce our experiments are available at https://github.com/Asafgendler/RSCP. Complete proofs of the theorems presented in this paper can be found in Supplementary Section S1. A complete description of the data sets used in our experiments, including pre-processing steps, can be found in Supplementary Section S2. The details about the models used in our experiments and how they were trained are provided in Supplementary sections S3 and S8.

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

# SUPPLEMENTARY MATERIAL: ADVERSARIALLY ROBUST CONFORMAL PREDICTION

## S1 SUPPLEMENTARY PROOFS

In this section, we prove Theorem 1 and Theorem 2 presented in Section 3 of the main manuscript.

*Proof of theorem 1.* From the definition of the prediction set in (8):

$$
\begin{aligned}
\mathbb{P}\left[Y_{n+1} \in \mathcal{C}_\delta(\tilde{X}_{n+1})\right] &= \mathbb{P}\left[\tilde{S}(\tilde{X}_{n+1}, Y_{n+1}) \leq Q_{1-\alpha}\left(\left\{\tilde{S}_i\right\}_{i \in \mathcal{I}_{\mathrm{cal}}}\right) + M_\delta\right] \\
\text{(using (7))} \quad &\geq \mathbb{P}\left[\tilde{S}\left(X_{n+1}, Y_{n+1}\right) + M_\delta \leq Q_{1-\alpha}\left(\left\{\tilde{S}_i\right\}_{i \in \mathcal{I}_{\mathrm{cal}}}\right) + M_\delta\right] \\
&= \mathbb{P}\left[\tilde{S}\left(X_{n+1}, Y_{n+1}\right) \leq Q_{1-\alpha}\left(\left\{\tilde{S}_i\right\}_{i \in \mathcal{I}_{\mathrm{cal}}}\right)\right] \\
&\geq 1 - \alpha,
\end{aligned}
$$

where the last inequality follows from the validity of the prediction set constructed by the vanilla split conformal procedure since $\tilde{S}(X_i, Y_i)$, $i \in \{\mathcal{I}_{\mathrm{cal}} \cup \{n+1\}\}$ are exchangeable (Vovk et al., 2005); see also (Romano et al., 2019, Lemma 2). □

*Proof of theorem 2.* From the definition of the prediction set in (5):

$$
\begin{aligned}
\mathbb{P}\left[Y_{n+1} \in \mathcal{C}(\tilde{X}_{n+1})\right] &= \mathbb{P}\left[\tilde{S}(\tilde{X}_{n+1}, Y_{n+1}) \leq Q_{1-\alpha}\left(\left\{\tilde{S}_i\right\}_{i \in \mathcal{I}_{\mathrm{cal}}}\right)\right] \\
\text{(using (7))} \quad &\geq \mathbb{P}\left[\tilde{S}\left(X_{n+1}, Y_{n+1}\right) + M_\delta \leq Q_{1-\alpha}\left(\left\{\tilde{S}_i\right\}_{i \in \mathcal{I}_{\mathrm{cal}}}\right)\right] \\
&= \mathbb{P}\left[\tilde{S}\left(X_{n+1}, Y_{n+1}\right) \leq Q_{1-\alpha}\left(\left\{\tilde{S}_i\right\}_{i \in \mathcal{I}_{\mathrm{cal}}}\right) - M_\delta\right] \\
\text{(using the definition of } \tau \text{, defined in (9))} \quad &\geq \mathbb{P}\left[\tilde{S}\left(X_{n+1}, Y_{n+1}\right) \leq Q_\tau\left(\left\{\tilde{S}_i\right\}_{i \in \mathcal{I}_{\mathrm{cal}}}\right)\right] \\
&\geq \tau,
\end{aligned}
$$

where, similarly to the proof of Theorem 1, the last inequality follows from (Romano et al., 2019, Lemma 2).

Lastly, we show that $\tau \leq 1 - \alpha$. From the definition of $\tau$ in (9):

$$
\begin{aligned}
\tau &= \max\left\{\tau' : Q_{\tau'}\left(\left\{\tilde{S}_i\right\}_{i \in \mathcal{I}_{\mathrm{cal}}}\right) \leq Q_{1-\alpha}\left(\left\{\tilde{S}_i\right\}_{i \in \mathcal{I}_{\mathrm{cal}}}\right) - M_\delta\right\} \\
&\leq \max\left\{\tau' : Q_{\tau'}\left(\left\{\tilde{S}_i\right\}_{i \in \mathcal{I}_{\mathrm{cal}}}\right) \leq Q_{1-\alpha}\left(\left\{\tilde{S}_i\right\}_{i \in \mathcal{I}_{\mathrm{cal}}}\right)\right\} \\
&= 1 - \alpha.
\end{aligned}
$$

□

## S2 DATA SETS

**Overview** We evaluate our methods on three widely used image classification data sets: CIFAR10, CIFAR100 (Krizhevsky, 2009), and ImageNet (Deng et al., 2009). The CIFAR10 data set consists of 60,000 RGB color images of size $32 \times 32 \times 3$, corresponding to 10 different classes, with 6,000 images per class. In this data set, there are 50,000 training images and 10,000 test images. The CIFAR100 data set is identical in structure to CIFAR10, except it has 100 classes containing 600 images each. Overall, this data consists of 50,000 training images and 10,000 test images. In our experiments, we use the entire training set for fitting the classifiers and split the rest 10,000 points into two sets of equal size (calibration and test) 50 times. We use the ILSVRC2012 version of the ImageNet data set. Here, the training set consists of 1.2 million color RGB images of different sizes, corresponding to 1,000 class labels. To construct calibration and test sets, we randomly split the 50,000 images of the ILSVRC2012 validation set into calibration and test sets of equal size, for 50 times. We use the entire ILSVRC2012 training set for fitting the predictive models.

**Pre-processing**   The CIFAR10 and CIFAR100 images are normalized by (i) subtracting the training images mean value, equals to [0.4914, 0.4822, 0.4465], and (ii ) dividing the result by the images standard deviation, equals to [0.2023, 0.1994, 0.2010]. The ImageNet images are first re-scaled to a size of $256 \times 256$. Then, the central $224 \times 224$ pixels are taken and normalized by subtracting the mean value of the training images [0.485, 0.456, 0.406] and dividing the result by their standard deviation, given by [0.229, 0.224, 0.225].

## S3   MODELS

As described in Section 5.1 of the main manuscript, for CIFAR10 and ImageNet data sets, we use the pre-trained ResNet-110 and ResNet-50 models from Cohen et al. (2019). We use the models that were trained on clean training points, as well as models that were trained on Gaussian augmented training points. These models are downloaded from `https://github.com/ locuslab/smoothing`. This link also provides the full details on how the models were trained.

Since the CIFAR100 data set was not studied by Cohen et al. (2019), we fit our own models by modifying the software package available at `https://github.com/bearpaw/ pytorch-classification`. Specifically, we fit a residual network of depth 110 (ResNet-110) by augmenting the training data with random crops of size 32 (after applying a zero padding of size 4) and random horizontal flips. The images are normalized as explained in Section S2. For convenience, we added the normalization as a pre-processing layer to the fitted model. As explained in Section 5.1 of the main manuscript, we fit two models: one on clean training points and another on points that are augmented with Gaussian noise of standard deviation $\sigma$ that is equal to the one used for smoothing. The models are trained using the stochastic gradient descent optimizer with a momentum term of 0.9 for 164 epochs, using a batch size of 128 points. The learning rate starts at 0.1 and is multiplied by a factor of 0.1 at epochs 81 and 122. A weight decay regularization of $10^{-4}$ is added to the loss function. We train the models using Pytorch, on a single Nvidia GEFORCE GTX 1080 Ti GPU.

## S4   WHY DOES TRAINING WITH GAUSSIAN NOISE IMPORTANT?

As stated in Section 4.2 of the main manuscript, while our method can work with any base classifier $\hat{f}$, in practice, we require $\hat{f}$ to have some degree of robustness in order to construct small prediction sets. Figure S6 demonstrates that our `RSCP` procedure constructs valid but very large prediction sets when deployed with a standard classifier, fitted without augmenting Gaussian noise to the CIFAR10 training points. Here, the average size of the prediction sets is equal to 9, a trivial result for the CIFAR10 data set that has only 10 classes. In this experiment, we use $\sigma = 0.5$ for smoothing, an attack of magnitude $\delta = 0.125$, and $n_s = 256$ Gaussian noise samples for estimating the smoothed score.

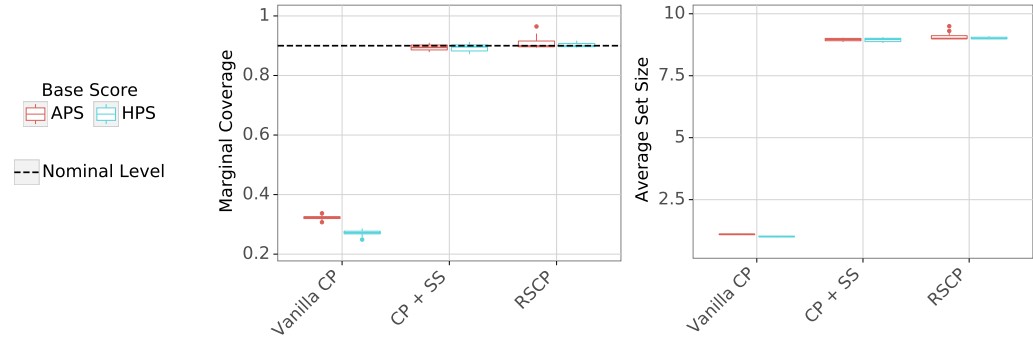

Figure S6: The importance of using robust models. Marginal coverage (left) and average set-size (right) obtained by different conformal methods. The target coverage level is 90%.

## S5 HYPER PARAMETERS

### S5.1 INFLUENCE OF THE VALUE OF $\sigma$

Herein, we extend the discussion from Section 5.1 of the main manuscript and study the effect of the choice of the hyper-parameter $\sigma$ used to compute the smoothed score. Specifically, we repeat the same experiment described in the context of Figure 3 for all data sets. Figure S7 shows how the ratio between $\sigma$ and $\delta$ affects the average set-size and the average coverage obtained by our RSCP method. In this experiment, we use an adversarial perturbation of magnitude $\delta = 0.125$ for CIFAR10 and CIFAR100, and magnitude $\delta = 0.25$ for ImageNet. As can be seen, a ratio of $\sigma/\delta = 2$ works well for all data sets and the two non-conformity scores.

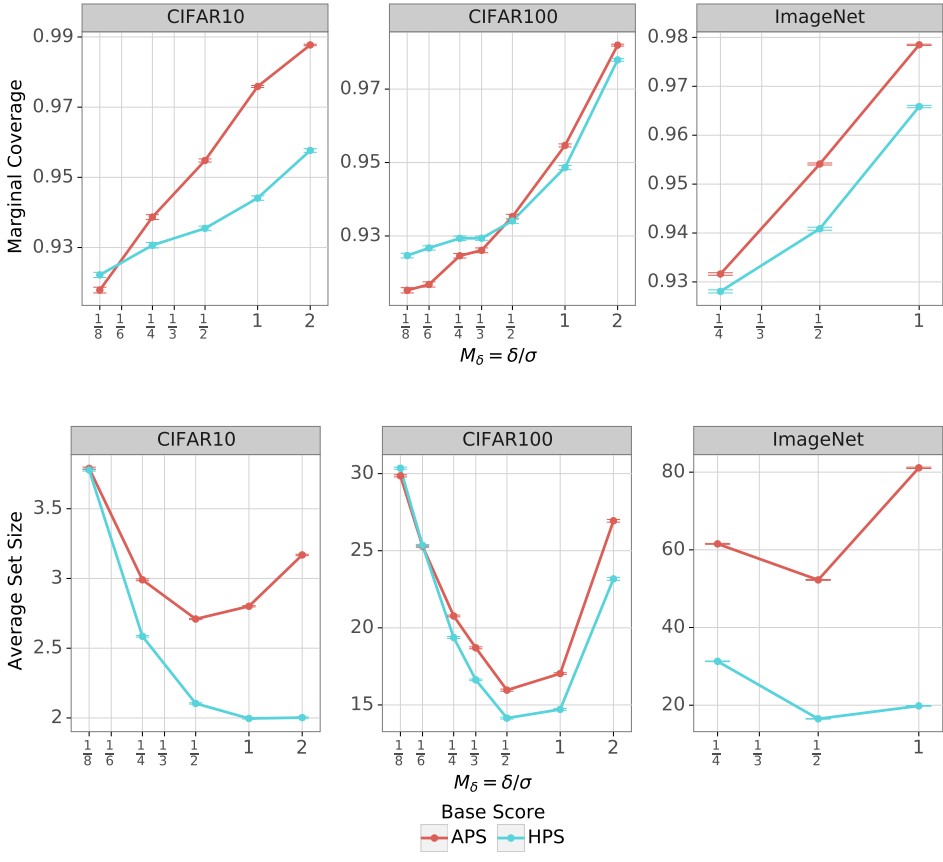

Figure S7: The effect of $\sigma$ on the marginal coverage (top) and size (bottom) of prediction sets constructed by RSCP. Here, $\delta$ and $n_s$ are fixed and equal to $0.125$ and $256$ for CIFAR10 and CIFAR100, and $0.25$ and $64$ for ImageNet.

### S5.2 INFLUENCE OF THE VALUE OF $n_s$

As explained in Section 5.1 of the main manuscript, the smoothed score is evaluated by replacing the expectation (10) with the empirical mean (12), averaging over $n_s$ i.i.d. Gaussian noise samples. Figure S8 shows the effect of the value of $n_s$ on the average set-size and the average coverage obtained by our RSCP method for all data sets. Following that figure, averaging more than $n_s = 256$ noise samples (or even $n_s = 64$) barely affects the empirical coverage and size of the constructed sets.

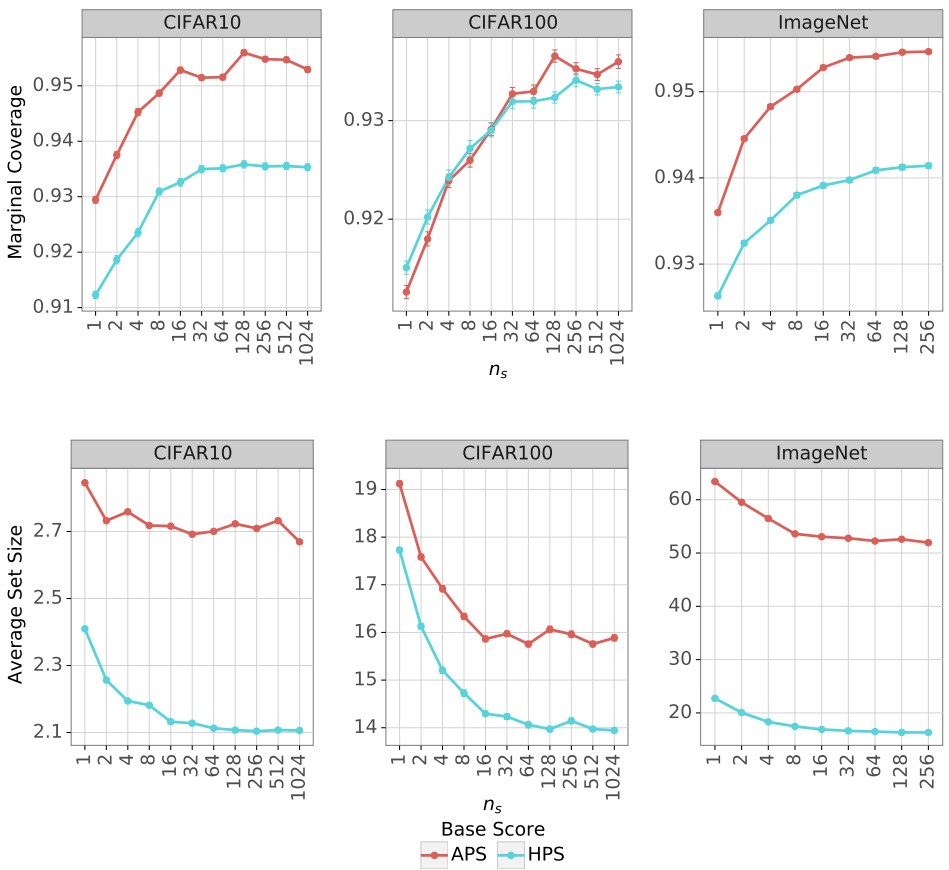

Figure S8: The effect of the number of noise realizations $n_s$ (used to estimate the smoothed score) on the marginal coverage (top) and size (bottom) of prediction sets constructed by RSCP. Here, $\delta$ and $\sigma$ are fixed and equal to $0.125$ and $0.25$ for CIFAR10 and CIFAR100, and to $0.25$ and $0.5$ for ImageNet respectively.

## S6    EFFECT OF THE ADVERSARIAL ATTACKS

As stated in Section 5.1 of the main manuscript, in all of our experiments, we use an attack of $\ell_2$ magnitude $\delta = 0.125$ for CIFAR10 and CIFAR100, and $\delta = 0.25$ for ImageNet. Table S1 illustrates the effect of such adversarial perturbations on the test accuracy of the models used in the experiments presented in Section 5 of the main manuscript as well as in Supplementary Section S5.1 and Supplementary Section S8 that is presented hereafter. The PGD attack leads to a significant reduction in the test accuracy of models that are fitted without Gaussian noise augmentation (i.e., $\sigma = 0$). By contrast, the models that are trained with Gaussian noise are more robust, as the decrease in test accuracy[3] is smaller when attacked by SMOOTHADV. This, however, comes at the expense of an accuracy reduction on the clean test points.

## S7    COMPUTATIONAL COMPLEXITY

Following Algorithm 1, at test time, the computational complexity of RSCP is dominated by the number of calls to the predictive model, required to compute the smoothed score for each of the $L$

---

[3]When evaluating the test accuracy of the models trained with Gaussian noise, we add Gaussian noise with the same standard deviation $\sigma$ to the test point since we find this approach to improve the accuracy.

Table S1: The influence of the adversarial attacks used in our experiments on the classifiers' test accuracy.

| Architecture | Training $\sigma$ | CIFAR10 | | CIFAR100 | | ImageNet | |
|---|---|---|---|---|---|---|---|
| | | Clean Test | Adversarial Test | Clean Test | Adversarial Test | Clean Test | Adversarial Test |
| ResNet | 0 | 89.94% | 26.7% | 71.03% | 12.19% | 75.69% | 19.59% |
| | 0.0625 | 87.13% | 68.96% | 64.58% | 41.17% | - | - |
| | 0.125 | 81.69% | 67.84% | 58.06% | 41.95% | - | - |
| | 0.25 | 75.19% | 65.18% | 50.73% | 40.82% | 68.58% | 56.32% |
| | 0.375 | - | - | 44.23% | 37.14% | - | - |
| | 0.5 | 63.84% | 57.34% | 39.0% | 33.68% | 61.04% | 53.09% |
| | 0.75 | - | - | 31.75% | 28.1% | - | - |
| | 1 | 47.99% | 44.41% | 25.57% | 23.56% | 47.73% | 42.89% |
| DenseNet | 0 | 95.42% | 23.28% | 77.07% | 4.29% | - | - |
| | 0.25 | 80.09% | 70.89% | 52.69% | 42.47% | - | - |
| VGG | 0 | 93.1% | 54.96% | 72.17% | 23.12% | - | - |
| | 0.25 | 78.88% | 69.82% | 48.57% | 39.54% | - | - |

possible labels $y \in \mathcal{Y}$. The number of calls to the predictive model for creating a set for a single test image is $n_s$, and the number of classes $L$ only affects the number of scores that need to be calculated using the model predictions. The training involves fitting a classifier, and the calibration requires computing the smoothed non-conformity scores for all calibration points, where both steps can be performed only once.

Table S2 summarizes the average runtime for constructing a prediction set using `Vanilla CP`, `CP+SS`, and `RSCP` with the HPS score function (3), for a single image, where the average is taken over 100 realizations. We present the results obtained by two different implementation strategies of our methods: 'single mode' and 'batch mode'. The 'single' implementation repeats the following two steps for $n_s$ times: (i) augment to the input image one realization of Gaussian noise, and (ii) feed this noisy image to the neural net model. The 'batch' implementation is more efficient since it parallelizes the inference step as follows: (i) augment $n_s$ noise realizations to the same input image, and (ii) pass all these noisy realizations together (as one batch) to the neural net model. We follow the exact same setup from the experiments used to create Figure 5 of the main manuscript. Specifically, $n_s = 256$ for CIFAR10 and CIFAR100, and $n_s = 64$ for ImageNet. The runtimes are evaluated using Pytorch, on a single Nvidia GEFORCE GTX 1080Ti GPU.

Following Table S2, we can see that when using the batch implementation the inference time of `CP+SS` and `RSCP` is approximately 3-4 times larger than `Vanilla CP`. Also, the runtimes of `CP+SS` and `RSCP` are almost identical, since the only difference between the two is the threshold used to construct the prediction sets.

Table S2: Inference time (seconds) for constructing a prediction set for a single image using different conformal prediction methods.

| Dataset | Vanilla CP | CP+SS | | RSCP | |
|---|---|---|---|---|---|
| | | Single | Batch | Single | Batch |
| CIFAR10 | 0.019336 | 4.820160 | 0.030831 | 4.920013 | 0.030856 |
| CIFAR100 | 0.017984 | 5.981254 | 0.030915 | 5.937026 | 0.030892 |
| ImageNet | 0.097539 | 1.288073 | 0.424692 | 1.134275 | 0.435865 |

## S8 RESULTS WITH DENSNET AND VGG MODELS

In this section, we study the performance of our method when applied with DenseNet (Iandola et al., 2014) and VGG (Simonyan & Zisserman, 2014) models on the CIFAR10 and CIFAR100 data sets,

as opposed to experiments presented thus far that use only ResNet (He et al., 2016) classifiers. The experiments here follow the protocol described in Section 5 of the main manuscript. We use the same attack algorithms and magnitude $\delta = 0.125$ as in the ResNet-110 experiments for CIFAR10 and CIFAR100. As for the hyper-parameters of our methods, we choose $\sigma = 0.25$ for smoothing, and $n_s = 256$ for estimating the smoothed score. Figure S9 compares the coverage and size of the prediction sets constructed by combining ResNet-110, VGG, or DenseNet with `Vanilla CP`, `CP+SS`, and `RSCP`. As can be seen, the difference between the three models is minor in terms of coverage for all calibration methods in both data sets. We can also see that our theoretical guarantees hold: for `CP+SS`, the worst coverage bound is below the empirical coverage obtained, and `RSCP` yields valid prediction sets in contrast to the two other methods. In terms of size, the difference between ResNet and DenseNet is minor, while VGG constructs larger sets, especially on CIFAR100.

Figure 1 of the main manuscript presents the results obtained by applying the VGG models to the CIFAR10 data set, using the HPS score. In that figure, we additionally tested the vanilla conformal method on the clean test points, to show the validity of this method when applied to clean test data that satisfy the exchangeability assumption.

Turning to the technical details about the training procedure. For each architecture, we fit two models: one on clean training points and the second on points that are augmented with Gaussian noise, as explained in Section 5.1 and Supplementary Section S3. All the models are trained on our local server. The DenseNet architecture consists of 100 layers with a growth rate of 12 and a compression rate of 2. We use the same data augmentation and normalization described in Supplementary sections S2 and S3. The models are trained using stochastic gradient descent with a momentum term that equals 0.9, a batch size of 64, and a total of 300 epochs. The learning rate starts at 0.1 and is multiplied by a factor of 0.1 at epochs 150 and 225. We also use a weight decay regularization that equals $10^{-4}$. The VGG architecture consists of 19 layers with batch normalization. We use the same data augmentation and normalization from Supplementary sections S2 and S3. The VGG models are trained using stochastic gradient descent: a momentum of 0.9, batch size of 128, for a total of 164 epochs. We set the initial learning rate to be equal to 0.1 and multiplied by a factor of 0.1 at epochs 81 and 122. We also use a weight decay regularization, which equals $10^{-4}$. The ResNet models are trained as explained in Supplementary Section S3. We train all the models using Pytorch, on a single Nvidia GEFORCE GTX 1080 Ti GPU.

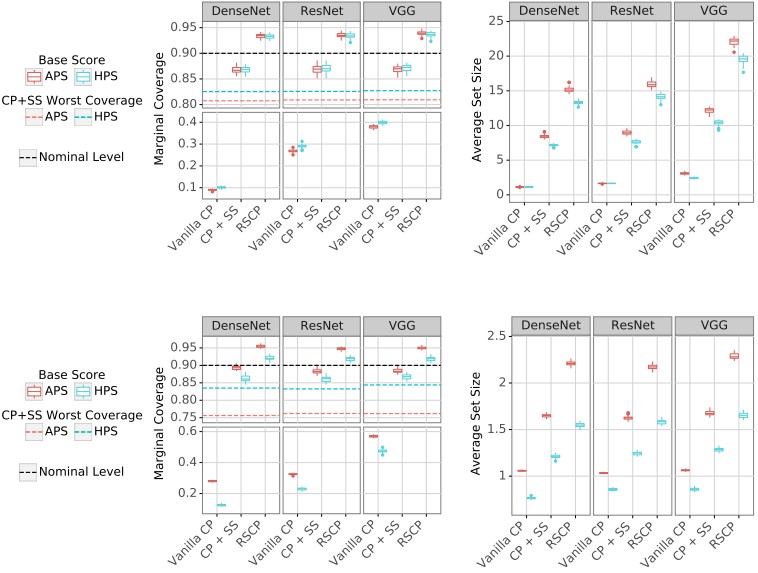

Figure S9: A comparison between ResNet, VGG, and DenseNet models, deployed to the CIFAR10 (bottom) and CIFAR100 (top) data sets. Marginal coverage (left) and average set-size (right) obtained by different conformal methods. The target coverage level is 90%.

## S9 RESULTS ON CLEAN DATA

In this section, we study the performance of our methods in a situation where no adversarial attack is performed. We follow the setup from Figure 5 of the main manuscript, and compare the three methods of interest (`Vanilla CP`, `CP+SS`, and `RSCP`) on the three data sets, using the same models, smoothing parameter $\sigma$, and the number of noise realizations $n_s$. Importantly, in contrast to Figure 5, the results that are presented in Figure S10 are obtained by applying these three methods on *clean test points* that are not corrupted by adversarial noise. Following that figure, both `Vanilla CP` and `CP+SS` achieve the desired coverage of $90\%$, which stands in line with the theoretical guarantee of split conformal prediction; see also Theorem 2. By contrast, `RSCP` results in a higher coverage rate than the desired one, since this method always accounts for the uncertainty induced by a possible attack even if not applied. Moving to the average set-size, notice that all the methods perform similarly to the case where the test data is corrupted, as depicted in Figure 5 of the main manuscript. Specifically, the average size of the sets constructed by `CP+SS` is larger than that of `Vanilla CP`, possibly due to smoothed model that is used to make the predictions. However, it is important to stress that `CP+SS` has a significantly smaller drop in coverage in the adversarial setting. We also note that the average size of the sets constructed by `RSCP` is larger than those of `CP+SS` due to the inflation of threshold $Q_{1-\alpha}$.

As a concluding remark, we iterate here the discussion from Section 3.2 of the main manuscript, displaying the two possible use-cases of our proposal. In the first, one can implement `RSCP` and construct valid prediction sets at any desired level of coverage, while bearing in mind that these sets are likely to be larger even if no attack is performed. In the second case, one can deploy `CP+SS`, guaranteeing that the sets will have the desired coverage in the case where *no attack is performed* while providing important information about the worst coverage that might be obtained under an adversarial attack. Moreover, we believe that in practice these two use-cases can be combined via the following two-step procedure to improve statistical efficiency: (i) apply a method for detecting whether a given test point is corrupted by an adversarial noise (e.g., Metzen et al., 2017; Feinman et al., 2017; Cohen et al., 2020); and (ii) if an attack is detected, apply the `RSCP` method to construct a valid prediction set, otherwise use the `vanilla CP` or `CP+SS` since these methods are less conservative.

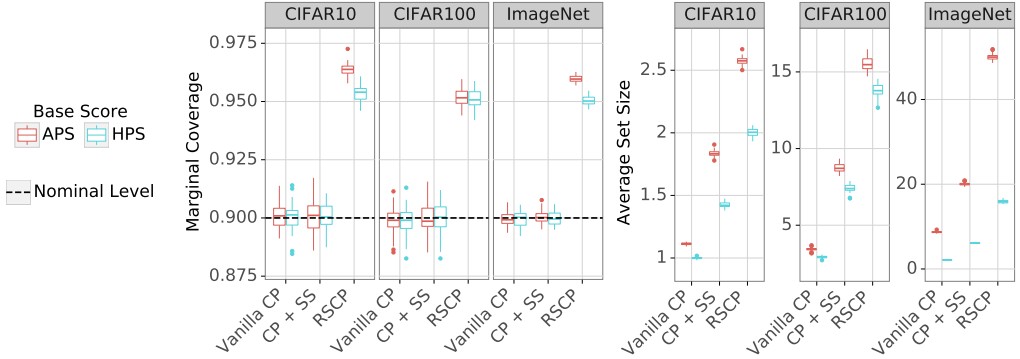

Figure S10: Performance on clean data. Marginal coverage (left) and average set-size (right) obtained by different conformal methods. The target coverage level is $90\%$.

## S10 RESULTS WITH AN ADVERSARIALLY-TRAINED MODEL

In this section, we study the performance of our methods, integrated with the adversarial training scheme of Salman et al. (2019) that was shown to improve the robustness to the specific `SMOOTHADV` attack we applied at test time. This new, more robust, predictive model replaces the one we used in Section 5 (see also Supplementary Section S3), which is fitted merely by adding Gaussian noise to the training points. Recall that the `SMOOTHADV` attack has two parameters: the magnitude of the noise $\delta$ and the smoothing strength $\sigma$. Here, we use a training procedure that is ideal in the sense that it has access to the same $\delta$ used for attacking the model, and the same $\sigma$ used

to compute the smoothed score. We perform this experiment only on the CIFAR10 data set and fit a ResNet-110 model using the code provided by Salman et al. (2019). Their software package is available at `https://github.com/Hadisalman/smoothing-adversarial`; this link also provides the full details on how the models are trained.

The results are depicted in Figure S11, which follows the exact same experimental protocol used to create Figure 5 of the main manuscript. That figure compares two versions of `CP+SS`: the first is implemented with a model fitted on Gaussian augmented data, and the second is implemented with the adversarially trained model of Salman et al. (2019). As can be seen, the latter model achieves a slightly better coverage, closer to the desired one with the HPS score, while achieving a slightly worse coverage with the APS score. For both scores, however, this model improves the statistical efficiency. Interestingly, even in this optimistic case, `CP+SS` does not attain the desired coverage perfectly. It is also important to stress that if the new model would be attacked by a different and more effective attack, the coverage can possibly drop even further. It is worth noting that this experiment supports our discussion from Section 6, arguing that any new development for improving adversarial robustness could be easily integrated with our method to improve the overall performance.

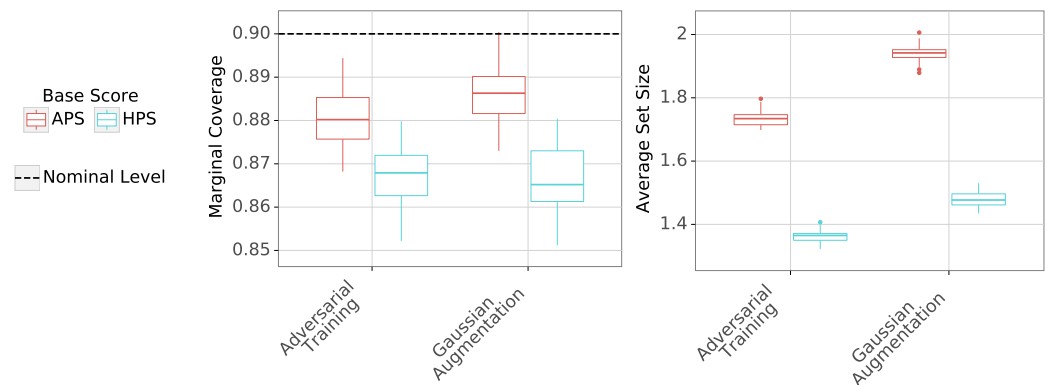

Figure S11: Results using an adversarially trained model. Marginal coverage (left) and average set-size (right) obtained by `CP+SS` on the CIFAR10 dataset with two different models. The target coverage level is $90\%$.

## S11 HPS vs. APS

In this section, we discuss the difference between the APS and HPS non-conformity scores and demonstrate the advantages that APS has over HPS. Romano et al. (2020b) carefully argued that while conformal prediction methods—which use the HPS score—are guaranteed to achieve the marginal coverage criterion (1), they may fail to achieve the stronger and more informative notion of conditional coverage, defined as

$$\mathbb{P}\left[Y_{n+1} \in \mathcal{C}\left(x\right)|X_{n+1}=x\right] \geq 1-\alpha. \tag{S14}$$

In plain words, the above statement requires the coverage to hold for a *specific observation* $X_{n+1} = x$, and not merely on average over all test points. While it is known that conditional coverage cannot be achieved in finite samples without imposing strong regularity conditions and/or modeling assumptions (Foygel Barber et al., 2021; Vovk, 2012), it was shown by Romano et al. (2020b) that the APS score performs better in terms of conditional coverage compared to the HPS score. In fact, Romano et al. (2020b) rigorously showed that by combining the APS score with an oracle classifier that has access to the true conditional class probabilities $P_{Y|X}$, one can exactly satisfy (S14).

To demonstrate the advantage of APS in the adversarial setting, we repeat the experiment from Figure 5 of the main manuscript and present the coverage and the average set-size, conditional on each of the 10 classes of the CIFAR10 data set. Following Figure S12, the APS prediction sets achieve a steady coverage across all classes, close to the desired level for the `CP+SS`, while the coverage rates of HPS vary greatly. Moreover, the latter method tends to undercover the labels of the

samples that belong to classes 2-5. One can also see that the size of the prediction sets constructed by APS varies across the different classes, and therefore better reflecting the prediction uncertainty. This stands in contrast with the sets constructed via the HPS score, whose sizes are more or less constant for all class labels.

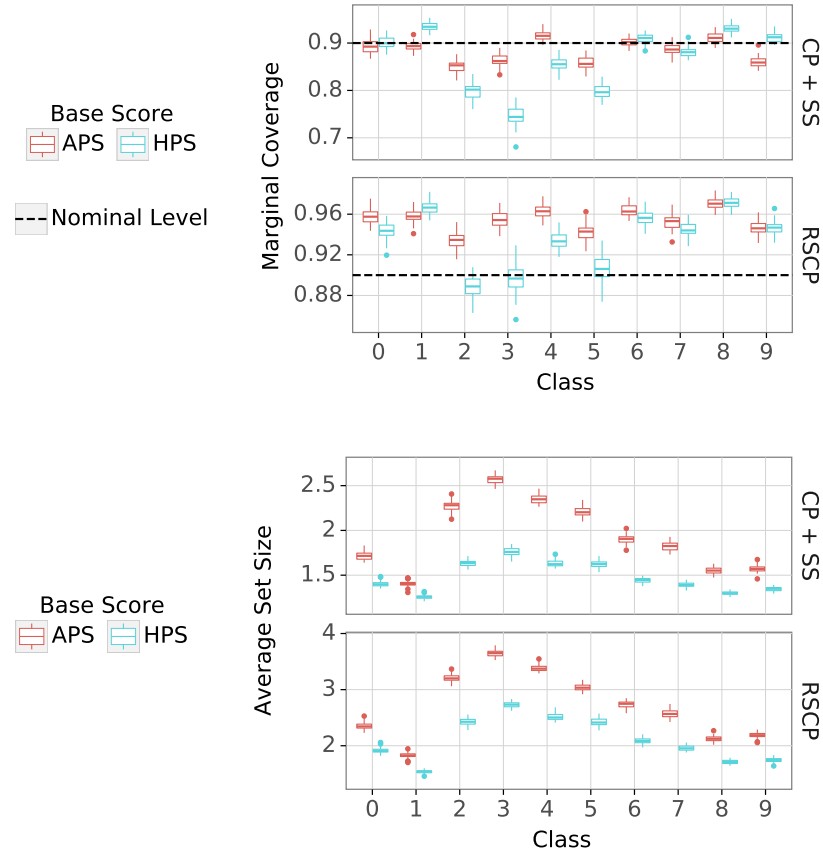

Figure S12: Comparison between HPS and APS. Marginal coverage (top) and average set-size (bottom) obtained by different conformal methods for each of the classes of CIFAR10 separately. The target coverage level is $90\%$.

## S12   COMPARISON WITH CAUCHOIS ET AL. (2020)

As discussed in Section 4, the task of reliably handling distribution shifts between the calibration and the test points has been studied in the literature of conformal prediction. However, none of the existing methods addresses our special setup, for which a full distributional shift is induced by an adversarial attack. Perhaps the closest work to ours is that of Cauchois et al. (2020). They present a calibration framework to construct robust prediction sets, given that the $f$-divergence between the test distribution and the calibration distribution of the non-conformity scores is bounded by a fixed value $\rho$. Since $\rho$ is unknown in practice, they propose a method to estimate it from the data and use this approximation to inflate the threshold $Q_{1-\alpha}$, so that the constructed sets will be robust to 'natural' distributional shifts. This is in contrast to our method that assumes that an unknown additive adversarial noise is added to the test points, and, as such, it is impossible to estimate $\rho$ in this case from the training/calibration data.

To demonstrate this, we follow the experimental protocol from Section 5 to compare our methods with Cauchois et al. (2020). We estimate $\rho$ by applying Cauchois et al. (2020, Algorithm 1), and then construct prediction sets for the test points that are corrupted by adversarial perturbations of

magnitude $\delta = 0.125$.[4] We use the same attacks detailed in Section 5, and evaluate this method over 50 random splits of the CIFAR10 dataset. We compare the marginal coverage and average set sizes achieved by their method to our `RSCP` approach with $\sigma$ matching to the one used to augment the training points, and set $n_s = 256$. We make this comparison with four different ResNet models: one that is trained on clean data (in this case we perform `RSCP` with $\sigma = 0.25$ for smoothing), and three that are trained on points augmented with Gaussian noise, each one with a different $\sigma$ [5].

The results are depicted in Figure S13. As can be seen, while our method achieves the desired coverage, no matter which base model is used (at the expense of larger prediction sets), the method of Cauchois et al. (2020) fails when using a model trained on clean points: the empirical coverage is below the desired coverage (Figure S13a). The same happens with the $\sigma = 0.0625$ and $\sigma = 0.125$ models; see figures S13b and S13c. When using a more robust classifier—trained with $\sigma = 0.25$— Cauchois et al. (2020) achieves the desired coverage in practice, although it is not supported by a theoretical guarantee; see Figure S13d. Interestingly, for this model, while the two methods construct prediction sets that achieve the desired coverage, our `RSCP` method—which uses smoothed scores—constructs smaller sets with the choice of HPS as the base score, and the same size of sets with the APS score.

---

[4]We thank the authors of (Cauchois et al., 2020) for sharing their code with us.

[5]As stated in Section S6, we add Gaussian noise to the test points when applying these models to calculate the non-conformity scores.

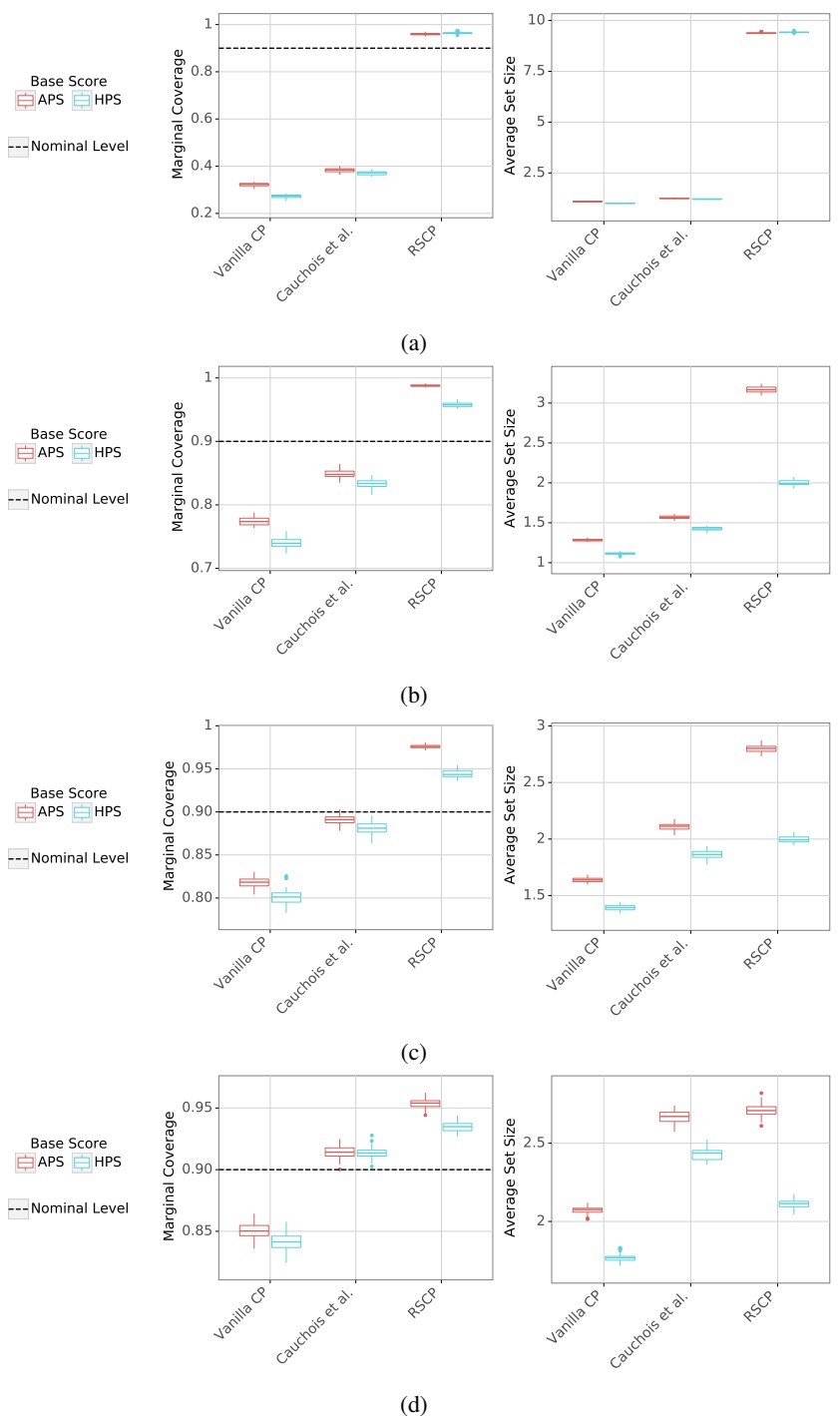

Figure S13: Comparison with Cauchois et al. (2020). Marginal coverage (left) and average set-size (right) obtained by different conformal methods on the CIFAR10 dataset with three different ResNet-110 models. The target coverage level is $90\%$. (a) model trained on clean training points, (b) model trained on points augmented by Gaussian noise with $\sigma = 0.0625$, (c) model trained on points augmented by Gaussian noise with $\sigma = 0.125$, (d) model trained on points augmented by Gaussian noise with $\sigma = 0.25$.

