# OpenReview forum: "Adversarially Robust Conformal Prediction"
_ICLR.cc/2022/Conference — ICLR 2022 Poster_

### Official Review · Reviewer_m5fr · 2021-10-26

**Correctness:** 3
**Technical Novelty And Significance:** 2
**Empirical Novelty And Significance:** 2
**Recommendation:** 5
**Confidence:** 4

**Main Review:**

**Strengths**:
1. This paper considers an under-explored problem, i.e., conformal prediction under adversarial perturbations.
2. The paper is easy to follow.

**Concerns**:
1. (novelty on theoretical guarantee) The novelty on the theoretical result is less significant; the correctness of the main theorem is heavily dependent on the condition of a non-conformity score in (7). In particular, finding the Lipschitz constant M_\delta is the challenging problem and actively researching area (as the paper pointed out), especially the score function is a highly non-linear neural network and an input is high-dimensional. However, given the score function that satisfies (7), constructing a conformal prediction set is relatively straightforward (even though I agree that it's a new result).

2. (preciseness on theoretical guarantee) The statement of Corollary 1 needs to be described more precisely, considering the fact that the literatures of conformal prediction are trying to be rigorous on making the theoretical guarantee; Corollary 1 requires randomly smoothed score function, but we cannot evaluate this function due to the expectation without approximation (as described in the paper). In this case, it would be more appropriate to say "the prediction set is asymptotically valid" or consider the samples required for approximating the expectation as a part of sample complexity analysis for the proposed prediction set. I think the latter making this paper theoretically more interesting and novel.

3. (comparison) The most closely related work is Cauchois et al. (2020), but the comparison is missing. The related work section points out the limitation of this work, i.e., "the f-divergence measure is notoriously difficult to estimate in practice"); however the proposed approach of Cauchois et al. (2020) is actually evaluated on CIFAR10 and ImageNet; in particular, the core part of this approach is convex optimization, so could be computationally not expensive. Finally, the results of Cauchois et al. (2020) is similar to the results of this paper, so the comparison looks necessary.

4. (CP+SS v.s. RSCP) I think it's not easy to say that RSCP is better than CP+SS; I understand that RSCP satisfies the coverage criterion, but that criterion is satisfied only for the next single example in a usual setup---the probability in (1) is taken over all calibration examples and (X_{n+1} , Y_{n+1}). If the constructed prediction set is conditioned on one fixed calibration set and evaluated over multiple testing examples, the empirical coverage probability of the standard conformal prediction is around the nominal level \alpha (see Figure 2 in Tibshirani et al. (2019)). In this sense, it's unclear whether RSCP is better than CP+SS. If the paper wants to claim that RSCP is better than CP+SS, I think PAC-style prediction set definition is required (e.g., [R1], [R2], [R3], or [R4]), which construct a prediction set conditioned on a fixed calibration set for correctness guarantee.

* [R1] https://arxiv.org/abs/1209.2673
* [R2] https://arxiv.org/abs/2001.00106
* [R3] https://arxiv.org/abs/2101.02703
* [R4] https://arxiv.org/abs/2106.09848


5. Other comments
- The description on conformal prediction is not precise; the paper said it requires i.i.d. but exchangeability is enough. Moreover, it can be applied to regression.
- Related to the above, is the i.i.d. assumption in Theorem 1 necessary? Is the proposed approach only applicable to classification?
- "however, none of them addresses the adversarial setting specifically" and "Both approaches differ from ours, which handles a full distributional shift induced by the adversarial perturbation": I think they are strong or incorrect since the adversarial setting is a special case of covariate shifts.



**Summary Of The Paper:**

This paper tackles the conformal prediction problem under adversarial perturbations, where the conventional exchangeability is violated. The proposed approach combines the non-conformity score with randomized smoothing and the standard conformal prediction. The proposed approach is evaluated over three different image benchmarks (i.e., CIFAR10, CIFAR100, and ImageNet) and demonstrated its efficacy by comparing the naive conformal prediction and the naive conformal prediction with the randomly smoothed non-conformity score.

**Summary Of The Review:**

Overall, this paper handles an interesting and timely problem, but the theoretical and empirical results are less significant compared to the known literatures as described in the main review; I lean to reject, but willing to discuss to adjust my understanding.

---

> ### Author Response · Authors · 2021-11-15
> **Response to Reviewer m5fr**
>
> We thank the reviewer for all the comments and for putting the time and effort into improving our manuscript. We carefully responded to the reviewer's criticisms point-by-point, in a response document that we have attached to the "Supplementary Material" section of our revised submission.
>
> **Please download the .zip file from the Supplementary Material and open the *response.pdf* file to view our response.**
>
> We will also include a major revision of our manuscript until the end of the discussion period, and a file that will include a marked-up version of our submission to track the changes easily.
>
> ==== EDIT =====
>
> **We uploaded a revised version of our manuscript, replacing the original submission. We also included at the end of the *response.pdf* file, a marked-up version of our submission to track the changes easily. Please download the *response.pdf* file from the Supplementary Material .zip, presented above.**
>
> Unfortunately, we have not included the comparison to the method of Cauchois et al. (2020) yet. This is because the software package the authors shared with us requires several major modifications to handle our setting. We are implementing these changes and will include the required comparison in the revised version of our manuscript, only after the rebuttal deadline.
>
> ==============

---

> > ### Comment · Reviewer_m5fr · 2021-11-28
> > **Discussion**
> >
> > Thanks for the clarification; I think my concerns still remain, and the following includes the additional discussion for it.
> >
> > 1. (covariate shift) The covariate shift assumes the labeling function p(y|x) does not change after the shift. This labeling function is independent to covariate distribution p(x), meaning that even though the adversary changes the covariate distribution, the labeling function does not; thus the authors' response that the adversary can change the labeling function is not correct. Note that to consider the adversarial setup as the distribution shift, the adversary needs to have the capability on changing the labeling function directly, but the labeling function is related to "common sense" on the label distribution of an example, which an adversary cannot change. Thus, I still think this paper assumes the covariate shift, and need to update the manuscript properly (as suggested before).
> >
> >
> > 2. (comparison) Partially related to the above, it's important to add comparison to Cauchois et al. (2020); it's unfortunate that the code is not ready, but we need to acknowledge the existing and directly related work.
> >
> >
> > 3. (CP+SS v.s. RSCP) For "but are performed by randomly splitting the data into disjoint calibration and test sets. Hence, those experiments are aligned with the setup of (1).", does this mean for each test example, we have a different calibration set to construct the prediction set? Only in this case, I think the experiments are aligned with the setup of (1) and the paper can claim that RSCP is guaranteed to satisfy (1). Alternatively, we can have a single calibration set, construct a prediction set using it, and evaluate all test examples using the same prediction set constructed using the same calibration set (and I call it an alternative setup). In this case, I would expect that the ideal result for RSCP is the box plot that is around the desired coverage level, similarly shown in Figure 2 of Tibshirani et al. (2019). However, the current results show RSCP is slightly above the desired coverage level and CP+SS is slightly below the level, so both approaches look good to me (again considering the marginal coverage of the "ideal" prediction set is around the coverage level in this alternative setting). In short, I'm not sure whether we can say CP+SS does not satisfy the guarantee (1) under the alternative setup, so I'm not sure whether RSCP is better than CP+SS.

---

> > > ### Author Response · Authors · 2021-11-29
> > > **Further Response to Reviewer m5fr**
> > >
> > > Thank you for your time and effort in reviewing our paper, which will certainly improve our manuscript.
> > >
> > > **Comment 1: (covariate shift)**
> > >
> > > Thank you for the opportunity to further clarify this point. We respectfully disagree with the reviewer, and we insist that an adversarial attack can induce a full distributional shift. We carefully designed two simple analytic examples that support our claim. We would really appreciate it if the reviewer could point out a specific problem that he/she finds in these examples. In essence, the assumption in the adversarial setting is that **the label remains the same after attacking the model,** rather than the conditional distribution of the labels given the features remains the same, as happens in covariate shift. As we demonstrated analytically by our examples, imposing this assumption can change both the distribution of the features (which is obvious) and the conditional distribution of the labels given the features. Perhaps a simple way to see this is that **the adversarial noise itself is a function of both the label and the features**. We understand that this issue is very confusing, perhaps ill-defined, and may even distract the reader. Therefore, we will seriously consider rephrasing "full distributional shift" with "distributional shift induced by the attacker."
> > >
> > >
> > > **Comment 2: (comparison)**
> > >
> > > We absolutely agree that a comparison with the method of Cauchois et al. (2020) will strengthen our paper. We will include this comparison in the revised paper. As we explained in the initial response, there are several major modifications that we are making to the software package the authors had recently shared with us.
> > >
> > > Nevertheless, we would like to emphasize the inherent problem in such a comparison. Cauchois et al.'s (2020) goal is to estimate from the calibration points the worst deviation in an f-divergence sense between the calibration and test points. This results in prediction sets that are robust against such deviation, which can be only estimated. Our framework, on the other hand, constructs prediction sets that are valid under a predefined $\ell_{2}$-bounded adversarial noise. Such a deviation **cannot be estimated from the calibration points since (i) it is not "a natural shift" (using the language of the authors), and (ii) it is not observed**. Hence, the estimation techniques proposed by Cauchois et al. (2020) (Algorithms 1-2) are not designed to handle the setting we study in our paper.
> > >
> > >
> > > **Comment 3: (CP+SS v.s. RSCP)**
> > >
> > > As a point of reference that the adversarial noise reduces the coverage of *CP+SS* we kindly refer the reviewer to the new experiments with clean data (without noise) we conducted; see Supplementary Section S9. There, the reviewer could see that the average coverage of *CP+SS* is 90% and not below this desired level, in contrast to Figure 5.
> > >
> > > We take this opportunity to clarify that the calibration set in our experiments is random, and not fixed. The reviewer may consider this approach to be a cheaper version of his/her suggestion to have a different calibration set for each test point (whose computational complexity is prohibitively large for the data sets we considered). Specifically, in our experiments, we repeat the following steps for 50 trials as a way to empirically evaluate Eq. (1). First, we **randomly** split the data that was not used for training into calibration and test sets. Second, we construct prediction sets for the test data using the threshold $Q_{1-\alpha}$ evaluated on the calibration set. Third, we compute the average coverage obtained on the test set. This process results in 50 points (average coverage per trial), which are used to form the box plots in Figure 5 (and also Figure S10). Importantly, this experimental protocol is very common in the literature on split conformal prediction and is used to test the validity of Eq. (1) presented in the paper. Following Figure 5, one can see that the **average** coverage of the prediction sets (average over the 50 trials) constructed by *CP+SS* is below the desired level, hence it does not satisfy Eq. (1) empirically (Again, this stands in contrast with Figure S10). On the other hand, the **average** coverage of the prediction sets constructed by *RSCP* is above the desired level, therefore satisfying Eq. (1) empirically. The most important point of this discussion is that *CP+SS* is not guaranteed to attain the desired coverage, in contrast to *RSCP*.

---

> > > > ### Comment · Reviewer_m5fr · 2021-11-30
> > > > **Additional Discussion**
> > > >
> > > > Thanks for the response; I clarified my descriptions in the following.
> > > >
> > > > **1. (covariate shift)**
> > > >
> > > > Let me put it this way; covariate shift is an assumption on the shift, which is not a target to be proven in general. So, it does not make sense to *prove* that the covariate shift assumption holds or not. Also, people tend to makes the covariate shift assumption to make the problem feasible (e.g., domain adaptation); without an assumption that p(y|x) = q(y|x), we cannot leverage the label information in a calibration set for getting a desired coverage after the shift. The paper can claim that it assumes a more general distribution shift, but it sounds self-contradictory---how can the labeled calibration examples be useful after the shift?
> > > >
> > > > **2. (comparison)**
> > > >
> > > > Assuming "deviation" means \rho in Cauchois et al.'s (2020), this paper does not estimate \rho; it is a hyperparameter as the proposed approach also has, i.e., \delta and \sigma. It's not easy to map \delta & \sigma into \rho, but there could be some \rho by which the prediction set by Cauchois et al.'s (2020) is robust to some adversarial perturbations, so comparable in some level.
> > > >
> > > > Anyway, the authors originally mentioned that they do not compare with Cauchois et al.'s (2020) since "the f-divergence measure is notoriously", but I don't think this is true. Cauchois et al.'s (2020) is not specifically tailored for adversarial perturbations, so the proposed approach may be better. However, the correct understanding on similarities and differences are clearly missing.
> > > >
> > > > **3. (CP+SS v.s. RSCP)**
> > > >
> > > > I would clarify a few points first.
> > > >
> > > > * "The reviewer may consider this approach to be a cheaper version of his/her suggestion to have a different calibration set for each test point (whose computational complexity is prohibitively large for the data sets we considered)": I do not suggest the new scheme; it is what is written in (1).
> > > > * " Importantly, this experimental protocol is very common in the literature on split conformal prediction and is used to test the validity of Eq. (1) presented in the paper. ": Yes, I understand this. That's why I mentioned Figure 2 of Tibshirani et al. (2019); the original scheme is computationally prohibitive, the prediction set needs to use the cheaper version of the original scheme for evaluation, and that's why the distribution of empirical coverage is around the desired coverage in Figure 2 of Tibshirani et al  (2019) (and also in Figure S9; thanks). Thus, this means it is hard to say that a method is bad when the empirical coverage box plot that is below or above the desired coverage.
> > > >
> > > > The following is the summary of my concerns under the "cheaper version" evaluation setup where CP with clean data anyway does not *strictly* satisfy (1):
> > > > * I think the empirical coverage distribution of the "ideal" RSCP is also around the desired coverage; but the RSCP in the paper is more conservative than this due to the loose bound in (11). I think this unfortunately conservative result is fortunately making the RSCP results looks *strictly* satisfying (1), which is misleading (under the "cheaper version" evaluation setup).
> > > > * We cannot say CP+SS is bad since it does not *strictly* satisfy (1); anyway it's slightly below of the desired coverage level. I'd say CP+SS is as good as RSCP---people may prefer CP+SS since its empirical coverage on clean data is around the desired coverage (which means it is efficient).
> > > >
> > > > Hope these comments help to improve the paper.

---

> > > > > ### Author Response · Authors · 2021-11-30
> > > > > **Response to Reviewer m5fr Additional Discussion**
> > > > >
> > > > > We thank the reviewer for the follow-up.
> > > > >
> > > > >
> > > > > 1. **" How can the labeled calibration examples be useful after the shift? "** \
> > > > > This is the question that we address in our paper for the specific adversarial setting we study. In other words, this is why we believe our contribution is important to the community.
> > > > >
> > > > >
> > > > > 2. **(comparison)** \
> > > > > Cauchois et al. (2020) (https://arxiv.org/abs/2008.04267) suggested a way to build valid coverage prediction sets if the f-divergence between the test distribution and the calibration distribution of the scores remains below $\rho$. In Section 3, they followed by claiming that "while the results in the previous section apply for a fixed shift amount $\rho$, a fundamental challenge is—given a validation data set—to determine the amount of shift against which to
> > > > > protect". Hence, they suggested algorithms 1 and 2 as a way to choose the value $\rho$ to protect against. In their empirical experiments on CIFAR10 and ImageNet in Section 4.3, they do not predefine $\rho$ by advance, instead, they use algorithms 1 and 2 to choose its value.\
> > > > > We absolutely agree that there “could be some $\rho$ by which the prediction set by Cauchois et al. (2020) is robust to some adversarial perturbations,” but it is not clear how to choose this value. We are happy that the reviewer agrees that mapping $\delta$ into $\rho$ is not an easy task. As we already mentioned, we will compare the performance of the two methods; we will make sure to cover the similarities and differences between the two methods as well.
> > > > >
> > > > >
> > > > > 3. **(CP+SS v.s. RSCP)** \
> > > > > Note that the statement $\mathbb{P}[Y \in C(X)] = \mathbb{E}[\mathbb{1}_{Y\in C(X))}]\geq 1-\alpha$ asks the average coverage to be above $1-\alpha$. However, the variance of the coverage may be large. In fact, the variance can be determined by the size of the calibration set, where a larger calibration set will reduce this variance. We kindly refer the reviewer to Section 3.1 in Bates et al. (2021) (https://arxiv.org/abs/2104.08279) for more details. In our statements, we refer only to the average coverage (this is the quantity we aim to control rigorously) and present the box plot to communicate the variance with the reader.\
> > > > >  Importantly, the **average** coverage of *CP+SS* is slightly below the desired level for the *specific attack and data sets we used* (in contrast to the clean case in which the **average** coverage is almost exactly the desired one). This brings us to the two use-cases we discussed for our proposal immediately after presenting Corollary 1 and also in Supplementary Section S9. We agree that some users may prefer *CP+SS* since its empirical coverage on clean data is guaranteed to be valid, and Theorem 2 reveals the worst-case coverage under the adversarial setting.\
> > > > > Moreover, the fact that "the *RSCP* results looks strictly satisfying (1)" is not "fortunate", using the reviewer's word, but exactly the opposite as it makes our method look over-conservative. Ideally, we would have wanted the **average** coverage of *RSCP* to be exactly the desired level and not above it. The importance of *RSCP* though, is that no matter how efficient new attacks will be in the future, its **average** coverage will never drop below the desired coverage as guaranteed by Theorem 1.
> > > > >
> > > > > We sincerely thank the reviewer for raising these points, which will improve our manuscript.

---

### Official Review · Reviewer_t8Jp · 2021-10-31

**Correctness:** 4
**Technical Novelty And Significance:** 3
**Empirical Novelty And Significance:** 3
**Recommendation:** 8
**Confidence:** 3

**Main Review:**

I like the proposed idea of using randomized smoothing to construct a non-conformity score that forms a prediction set with a finite-sample coverage guarantee under adversarial noise on the features. The paper is also written in a clear way which gives enough background knowledge to readers who are not familiar with the field of conformal prediction. The motivation of introducing randomized smoothing into the score function was very well illustrated, and the theorems 1 and 2 provided convincing results of the effectiveness of the proposed approach.

I have one major concern of the experimental section. Since the comparison was only done with the non-robust conformal prediction approach, I am wondering if it is possible to compare to other robust CP methods such as those mentioned in Section 4.1. In particular, the authors mentioned that this work "handles a full distributional shift induced by the adversarial perturbation". But my understanding is that this work considers only the perturbation on the feature X while P(Y|X) remains intact. It would be helpful if the authors can clarify on this point.

A minor point, there is a typo in Eq. (8), where it should be \tilde{S}(\tilde{X}_{n+1}, y).

**Summary Of The Paper:**

This paper generalized a data-splitting conformal prediction approach to the adversarial attack setting by combining conformal prediction with randomized smoothing, which yields a prediction set with finite sample coverage guarantee under an l2-norm bounded adversarial noise. The effectiveness of the proposed methods was demonstrated on the CIFAR10, CIFAR100, and ImageNet datasets, showing a significant increase in the coverage size compared to the vanilla conformal prediction method.

**Summary Of The Review:**

Overall I like the idea proposed in this paper. Their theoretical and experimental results demonstrated the effectiveness of the proposed approach. My major concern is about comparing with other robust conformal prediction methods in the experimental section. It would be helpful if the authors can provide insights on this.

---

> ### Author Response · Authors · 2021-11-15
> **Response to Reviewer t8Jp**
>
> We thank the reviewer for all the comments and for putting the time and effort into improving our manuscript. We carefully responded to the reviewer's criticisms point-by-point, in a response document that we have attached to the "Supplementary Material" section of our revised submission.
>
> **Please download the .zip file from the Supplementary Material and open the *response.pdf* file to view our response.**
>
> We will also include a major revision of our manuscript until the end of the discussion period, and a file that will include a marked-up version of our submission to track the changes easily.
>
> ==== EDIT =====
>
> **We uploaded a revised version of our manuscript, replacing the original submission. We also included at the end of the *response.pdf* file, a marked-up version of our submission to track the changes easily. Please download the *response.pdf* file from the Supplementary Material .zip, presented above.**
>
> Unfortunately, we have not included the comparison to the method of Cauchois et al. (2020) yet. This is because the software package the authors shared with us requires several major modifications to handle our setting. We are implementing these changes and will include the required comparison in the revised version of our manuscript, only after the rebuttal deadline.
>
> ==============

---

### Official Review · Reviewer_oSbg · 2021-11-01

**Correctness:** 3
**Technical Novelty And Significance:** 3
**Empirical Novelty And Significance:** 3
**Recommendation:** 6
**Confidence:** 4

**Main Review:**

The framework proposed in the work considers an important direction in the study of the applicability of conformal prediction in settings beyond the i.i.d. scenario. For safety-critical applications, it is important to build robust ways of quantifying uncertainty as certain failures could have disastrous consequences. The authors consider a popular setting where test data contains adversarial examples with bounded $\ell_2$-norm. For a modified procedure to work, the non-conformity scores have to satisfy a certain property, and the authors propose a way to incorporate any ``base'' score into the framework via randomized smoothing. I was wondering whether the authors could comment a bit more on the following questions:

--- Q1: While it is intuitive that getting robust prediction sets comes at the cost of larger prediction sets, I couldn't find a simulation that considers a null case. That is, a comparison of vanilla conformal against a robustified version proposed in this work in the setting where adversarial examples are not present, and thus i.i.d. assumption is sensible (i.e. vanilla conformal would have reasonable marginal coverage). Is it indeed the case that such simulation was not present in the paper? The reason is that one reasonable requirement for the prediction sets is their ``actionability'', i.e., it is hard to assess how actionable prediction sets of size 30 are in 100-classes classification problem.

--- Q2: Regarding computational complexity. It is clear that at the inference stage, for constructing a prediction set, it is necessary to perform sampling of perturbed input $n_S$ times for each class (as the authors point out in the Appendix S7). Are there any specific settings (i.e., pair of model architectures, datasets) that the authors have numbers for? It is interesting how computationally feasible the framework is.

--- Q3: I was wondering whether the authors can comment a bit more on the sensitivity of the smoothed scores to using approximations obtained via sampling. A partial answer is presented in S5.2 where some figures are presented for average set sizes and coverage. But it seems that if the original non-conformity scores are bounded, then it is possible to look at the sizes of the confidence intervals for the quantities that are being approximated.

--- A side note: equation 8 might contain a typo. it seems that $\tilde X_{n+1}$ (an observed feature vector) should be stated in place of $X_{n+1}$ (an unobserved one).

In general, the work is well-written equipped with important parts of (relevant) literature review and well-designed simulation studies.

**Summary Of The Paper:**

Quantifying predictive uncertainty is critical for various real-world applications of ML models. Post-hoc ``wrapper-style'' procedures for uncertainty quantification, which can be built on top of any black-box model and thus do not require modifications of training algorithms, are of great value due to a, typically, high number of engineering tweaks used during the model development stage. For example, (split) conformal prediction modifies an underlying point prediction model, which outputs the top-ranked label only, into a set-valued predictor that instead outputs a set of labels. Under the i.i.d. (or more generally, exchangeability) assumption, the resulting sets are provably valid (in terms of coverage) with guarantees being marginal over calibration and test data.

Violation of the i.i.d. assumption invalidates the inference, and adaptations to some structured distribution shifts have been proposed recently in the literature. The current work focuses on a setting where adversarial examples might be present at the test stage. Vanilla conformal is based on considering a collection of candidate prediction sets, parameterized by a single parameter, which is tuned/calibrated using a held-out set for performing set-valued predictions on test data. The authors propose correction of vanilla conformal, which essentially boils down to inflating the threshold in a way that guarantees robustness against adversarial examples with bounded $\ell_2$-norm (using randomized smoothing), where inflation results in outputting larger the prediction sets.

**Summary Of The Review:**

The current work represents a solid piece of work that takes a step forward towards robust conformal prediction. Within the context of CP under distribution shifts, prior works studied settings where structured/constrained shifts are present. This work focused on a different setting which hasn't been covered in conformal literature before where at the test stage adversarial examples invalidate the vanilla CP. The reasons for lowering the score include: limited theoretical contributions, a limited study of the types of adversarial examples, unaddressed questions stated in the main review.

**Update after rebuttal**

I would like to thank the authors for the detailed responses. Taking into account the general contribution of this work and points mentioned in this and other reviews, I tend to keep the current score.

---

> ### Author Response · Authors · 2021-11-15
> **Response to Reviewer oSbg**
>
> We thank the reviewer for all the comments and for putting the time and effort into improving our manuscript. We carefully responded to the reviewer's criticisms point-by-point, in a response document that we have attached to the "Supplementary Material" section of our revised submission.
>
> **Please download the .zip file from the Supplementary Material and open the *response.pdf* file to view our response.**
>
> We will also include a major revision of our manuscript until the end of the discussion period, and a file that will include a marked-up version of our submission to track the changes easily.
>
> ==== EDIT =====
>
> **We uploaded a revised version of our manuscript, replacing the original submission. We also included at the end of the *response.pdf* file, a marked-up version of our submission to track the changes easily. Please download the *response.pdf* file from the Supplementary Material .zip, presented above.**
>
> ==============

---

### Official Review · Reviewer_Cuc8 · 2021-11-05

**Correctness:** 4
**Technical Novelty And Significance:** 3
**Empirical Novelty And Significance:** 3
**Recommendation:** 8
**Confidence:** 4

**Main Review:**

The paper considers the conformal prediction problem in an adversarial setting, which is new and important in my perspective. It is very well-written, which I enjoyed reading. The idea of using randomized smoothing to construct a robust conformity score is quite novel and theoretically sound for conformal prediction, despite the techniques it used are well-known in the field of adversarial robustness. The empirical studies clearly demonstrate the vulnerability of standard conformal prediction method in the presence of adversarial examples, which support the motivation of the paper. Nevertheless, I have the following general questions for the authors:

1. In Figure 5, you show the marginal coverage against adversarially-perturbed inputs for different methods. How about the marginal coverage for normal examples?

2. The method “CP + SS” is not able to satisfy the desired coverage. Is this because the base classifier is trained to be robust against Gaussian noise instead of against worst-case adversarial perturbations? I am wondering whether you could achieve the coverage by using the conformity scores outputted by an adversarially-robust classifier?

3. One of the evaluation criteria for conformal prediction you used is average set size. Based on Figure 5, it seems that HPS based method outperforms APS based method by a large margin. But you mentioned in Section 2 that “APS reflect better the underlying uncertainty across sub-populations”. Do you have any empirical results supporting this? In general, how to decide which method to use if we want to deploy the conformal predictors you proposed.


**Summary Of The Paper:**

This paper proposes a generic method to construct conformal prediction sets in an adversarial setting. Since standard conformal prediction method assumes an i.i.d. assumption for training and testing input, its generated prediction sets for adversarial examples will not satisfy the coverage guarantee. Build upon on randomized smoothing, the paper then proposes a new non-conformity score and raise the threshold to account for the adversarial transformations. It then proves that the prediction sets constructed by the proposed method satisfy the coverage guarantee for worst-case scenarios. Empirical evaluations are also performed on benchmark datasets, which justify the effectiveness of their method.


**Summary Of The Review:**

Overall, I think the paper is well-written and reach the acceptance bar of ICLR. Both theoretical and empirical results support the main claim of the paper. Therefore, I vote an accept for this paper.

---

> ### Author Response · Authors · 2021-11-15
> **Response to Reviewer Cuc8**
>
> We thank the reviewer for all the comments and for putting the time and effort into improving our manuscript. We carefully responded to the reviewer's criticisms point-by-point, in a response document that we have attached to the "Supplementary Material" section of our revised submission.
>
> **Please download the .zip file from the Supplementary Material and open the *response.pdf* file to view our response.**
>
> We will also include a major revision of our manuscript until the end of the discussion period, and a file that will include a marked-up version of our submission to track the changes easily.
>
> ==== EDIT =====
>
> **We uploaded a revised version of our manuscript, replacing the original submission. We also included at the end of the *response.pdf* file, a marked-up version of our submission to track the changes easily. Please download the *response.pdf* file from the Supplementary Material .zip, presented above.**
>
> ==============

---

### Author Response · Authors · 2021-11-22
**General response to reviewers and submission guidelines**

We thank the four reviewers for all the comments and for putting the time and effort into improving our manuscript.

We uploaded a revised version of our manuscript, replacing the original submission (the changes are not marked up in this version).

We carefully responded to the reviewers' criticisms point-by-point, in a response document called *response.pdf*. Please download the *response.pdf* file from the Supplementary Material .zip, presented above and look at its first part to find our responses.

We also included at the end of the *response.pdf* file, a marked-up version of our submission to track the changes easily. Please download the *response.pdf* file from the Supplementary Material .zip, presented above and look at its second part to find the marked-up version.

The Supplementary Material .zip also contains the code for running all the experiments and creating all the figures in this manuscript under the *Project_RSCP* folder, accompanied by a readme.txt file that explains how to do it.

---

### Author Response · Authors · 2021-11-27
**Reminder**


Dear Reviewers,

Thank you for your efforts in reviewing our paper.
We kindly remind you that the discussion period ends in two days. We would sincerely appreciate any follow-up on our response. This way, we will know that you have
seen it, and we will be able to address any further concerns or questions that you may have.


Thank you,

The Authors

---

### Decision · Program_Chairs · 2022-01-20

**Decision:**

Accept (Poster)

**Comment:**

This paper studies the problem of producing distribution-free prediction sets using conformal prediction that are robust to test-time adversarial perturbations of the input data. The authors point out that these perturbations could be label and covariate dependent, and hence different from covariate-shift handled in Tibshirani et al 19, the label-shift handled in Podkopaev and Ramdas 21, and the f-divergence shifts of Cauchois et al 2021.

The authors propose a relatively simple idea that has appeared in other literatures like optimization but appears to be new to the conformal literature: (i) use a smoothed (using Gaussian noise on X, and inverse Gaussian CDF) nonconformity score function, in order to control its Lipschitz constant, (ii) utilize a larger score cutoff than the standard 1-alpha quantile of calibration scores employed in conformal prediction. The observation that point (i) alone lends some robustness to adversarial perturbations of the data is interesting. As several experiments in the paper and responses to reviewers show, this comes at the (apparently necessary) price of larger prediction sets.

I read through all the comments and also the supplement. The authors have responded very well to all the reviewers questions/concerns, adding significant sections to their supplement as a result. Three reviewers are convinced, but one remaining reviewer requested additional experiments to compare with Cauchois et al (in addition to all the others already produced by the authors originally and in response to reviewers). However, the authors point out that the code in the aforementioned paper was not public, but they were able to privately get the code from the authors during the rebuttal period. At this point, I recommend acceptance of the paper even without those additional experiments, since it is not the authors' fault that the original code was not public. Nevertheless, I suggest to the authors that, if possible, they could add some comparisons to the camera-ready since they now have the code.

I congratulate the authors on a nice work, a very solid rebuttal, and also the astute reviewers on pointing out various aspects that could be improved.

Minor point for the authors (for the camera-ready): I would like to comment on the Rebuttal point 4.4 in the supplement, which then got further discussed in the thread. The reviewer points out four references [R1-R4]. I will add one more to the list [R5] https://arxiv.org/pdf/1905.10634.pdf (Kivaranovic et al, appeared in 2019, published in 2020). I think the literature reviews in this area are starting to be messy, and all authors need to do a better job. Clearly, the original paper of Vovk et al already establishes various types of conditional validity (and calls it PAC-style guarantee), produces guarantees that others in this area produce, and it appears that much recreation of the wheel is occurring. For eg, [R2, R4] do not cite [R5], despite [R5] appearing earlier and being published earlier, and having PAC-style guarantees and experiments with neural nets, etc. However, in turn, [R5] do not cite Vovk [R1], but [R2, R4] do cite [R1]. (And [R3] does not seem to be relevant to this discussion of conditional validity?) In any case, I am not sure any of these papers need citing since the current paper does not deal with conditional validity. If at all, just one sentence like "Conditional validity guarantees, of the styles suggested by Vovk [2012], would be an interesting avenue for future work".